# How to Auto-optimize Prompts for Domain Tasks? Adaptive Prompting and Reasoning through Evolutionary Domain Knowledge Adaptation

**Yang Zhao**    **Pu Wang**    **Hao Frank Yang** *
Johns Hopkins University
{yzhao229, pwang80, haofrankyang}@jhu.edu
Project page: https://miemieyanga.github.io/EGOPrompt/

## Abstract

Designing optimal prompts and reasoning processes for large language models (LLMs) on domain-specific tasks is both necessary and challenging in real-world applications. Determining how to integrate domain knowledge, enhance reasoning efficiency, and even provide domain experts with refined knowledge integration hints are particularly crucial yet unresolved tasks. In this research, we propose Evolutionary Graph Optimization for Prompting (EGO-Prompt), an automated framework to designing better prompts, efficient reasoning processes and providing enhanced causal-informed process. EGO-Prompt begins with a general prompt and fault-tolerant initial Semantic Causal Graph (SCG) descriptions, constructed by human experts, which is then automatically refined and optimized to guide LLM reasoning. Recognizing that expert-defined SCGs may be partial or imperfect and that their optimal integration varies across LLMs, EGO-Prompt integrates a novel causal-guided textual gradient process in two steps: first, generating nearly deterministic reasoning guidance from the SCG for each instance, and second, adapting the LLM to effectively utilize the guidance alongside the original input. The iterative optimization algorithm further refines both the SCG and the reasoning mechanism using textual gradients with ground-truth. We tested the framework on real-world public health, transportation and human behavior tasks. EGO-Prompt achieves 7.32%–12.61% higher F1 than cutting-edge methods, and allows small models to reach the performance of larger models at under 20% of the original cost. It also outputs a refined, domain-specific SCG that improves interpretability.

## 1 Introduction

Foundation models, particularly Large Language Models (LLMs), are increasingly being adapted for domain-specific tasks, providing reasoning and decision support across real-world applications, such as public health [1–3], transportation [4–7], medical treatment [8, 9], and robotics [10–12]. For these models to be used effectively in particular domains, additional task adaptations are usually necessary. Prompt engineering has emerged as the primary, flexible, and cost-effective method for this adaptation [4, 13–16]. In this process, domain experts generally incorporate specialized knowledge and priors into prompt design by structuring the prompts and excluding irrelevant information to enhance the reasoning process. However, experts can also inadvertently introduce assumptions, mechanisms, even biases in prompt engineering [17, 18]. Therefore, critical questions arise – **how can we optimize prompts and reasoning procedures, and discover better combinations that integrate structured domain knowledge for domain-specific tasks? Furthermore, how can we automate this prompt-and-reasoning optimization?**

---

*Corresponding author.

39th Conference on Neural Information Processing Systems (NeurIPS 2025).

Seeking the potentially better prompt and reasoning procedure for domain-specific tasks without fine-tuning involves three challenges: **1) Domain-knowledge Adaptation** involves organizing extensive domain knowledge in textual form to maximize task performance while minimizing bias [19, 20]. For example, in traffic crash modeling [4, 21], LLMs must consider a broad range of textual inputs, including driver attributes, vehicle characteristics, infrastructure conditions, environmental factors, and driving behaviors, etc. Although these elements are represented in text, factors such as word choice, linguistic description, level of detail, and paragraph organization can influence how effectively the model learns and reasons about the domain. **2) Optimal Domain-adaptive Reasoning** recognizes that even with high-quality textual descriptions, physical priors still play a crucial role. Domain-specific conditional distributions and causal graphs strongly influence the reasoning process and performance [22]. Leveraging these priors can further guide the reasoning procedure, resulting in a more efficient overall process [23]. Without explicit guidance, LLMs may overlook key domain relationships [19], leading to hallucinations or outputs lacking verifiable evidence. To bridge the gap between a model's general reasoning abilities and the domain-specific priors needed for robust inference, it is crucial to incorporate causal structures and other form of domain representations. **3) Task Evolutionary** then considers whether LLMs can continue refining their reasoning once domain knowledge and causal relations are effectively integrated. For instance, can active knowledge or ground truth data help a domain-adaptive prompt match or surpass state-of-the-art legacy models? If so, the resulting improved domain knowledge and causal relations can further support evolving domain knowledge and uncover hidden factors, thereby reinvigorating expert-driven research.

In response, a promising approach is to leverage graph-based structural domain knowledge to guide both prompt design and the reasoning process [24]. Graph data offers explicit representations of entities and their conditional relationships, capturing key domain features correlations, whether factual (e.g., alcohol involvement increases crash severity regardless of consumption level) or causal (e.g., how alcohol consumption, road-surface conditions, and other factors jointly affect crash severity). By integrating these explicit relational structures, LLMs can better connect textual descriptions to domain-specific causal dependencies, reducing the likelihood of overlooked relationships or unsupported conclusions. Recent work has explored knowledge and causal graphs to enhance LLM reasoning [24–26], often via Retrieval-Augmented Generation (RAG) [27–29], which retrieves relevant subgraphs or facts to guide the model during inference [24, 25]. While this line of research is promising, there remain critical challenges for adaptive reasoning:

1. **Limited and Incomplete Domain Knowledge and Graphs.** In real-world settings, domain experts often have access only to partial knowledge or graphs for prompt design and reasoning [30]. However, many methods assume the availability of fully curated knowledge or causal graphs [26, 31], which is an unrealistic and task-inflexible requirement (e.g., in RAG scenarios where no relevant knowledge graphs exist).

2. **Interpretability and Knowledge Refinement.** Information in external graph databases may be inaccurate, yet existing methods often treat it as a ground-truth reference, potentially leading to performance degradation [18, 32, 33].

3. **Fixed External Priors.** Many current methods depend on fixed, external graph databases that can lack coverage or fail to actively evolve with the domain, limiting their adaptability to emerging contexts or newly available information [24–26].

4. **Automated Evolution.** In existing methods, the interaction is typically one-way: external knowledge is used to strengthen LLM reasoning, but the model does not feed back corrections or enhancements to the experts. As a result, experts must invest additional effort to extract new insights from the model's outputs [24–26].

Given these constraints, effectively adapting LLMs to novel tasks often requires an active process of graph integration and evolution for improved prompt design and reasoning. In this research, we propose **EGO-Prompt** (Evolutionary Graph Optimization for Prompt) for evolutionarily incorporating domain structural knowledge into more effective reasoning in LLMs. As shown in Figure 1, we begin by representing expert knowledge as an active, expert-constructed Semantic Causal Graph (SCG) $\mathcal{G}$. EGO-Prompt then decomposes the graph-guided reasoning process into two stages: 1) Generating instance-specific reasoning guidance $z^*(x, \mathcal{G})$ derived from the SCG $\mathcal{G}$ and the input data $x$, subject to output constraints, and 2) Performing reasoning conditioned on this guidance. This process can be abstracted as the following equation:

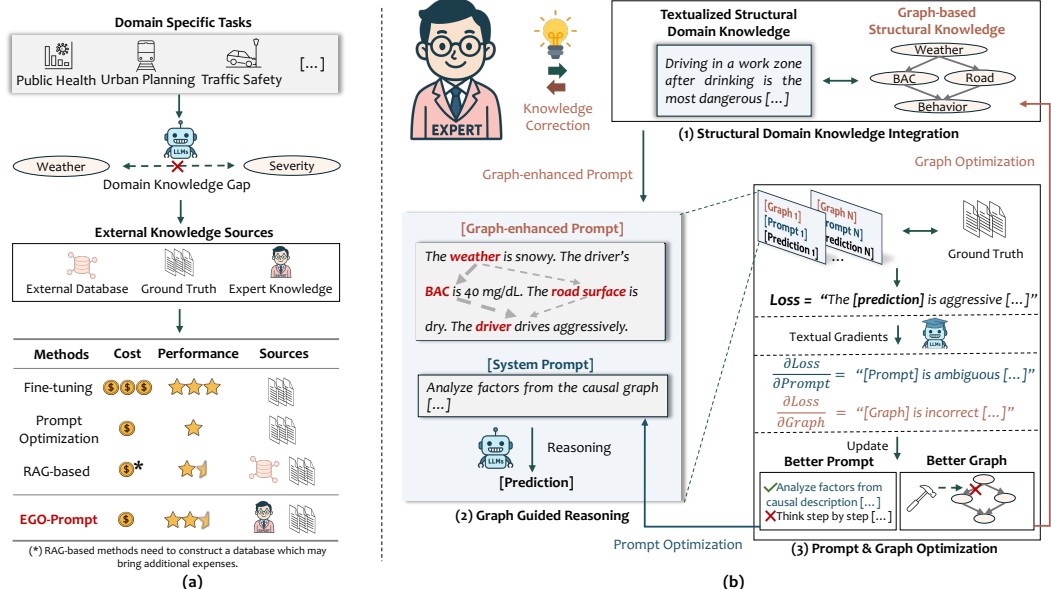

Figure 1: Overview of the proposed EGO-Prompt. (a) LLMs often struggle with domain-specific tasks due to the optimal prompt design and domain knowledge gap. Existing methods rely on the external database or established graph. In comparison, EGO-Prompt evolutionarily incorporates expert knowledge with minimal cost. (b) We represent external knowledge as a graph-based structure. A graph-enhanced prompt is then generated to guide the LLM's reasoning. Both the graph and the prompt are iteratively optimized using textual gradients from ground-truth data.

$$p(y \mid x, \mathcal{G}) \;=\; p\big(y \mid x, \overbrace{z^*(x, \mathcal{G})}^{\text{Instance-specific Reasoning Guidance}}\big) \tag{1}$$

where $y$ is the target reasoning result. The derivation is provided in Appendix 8.1. With this reasoning workflow, both the SCG and the reasoning process are then jointly refined through an evolutionary optimization algorithm that learns factual patterns from ground-truth data. This dynamic adaptation not only enhances the model's reasoning accuracy but also improves the quality of the SCG and its alignment with the prompt. Experiments across three domain-specific tasks demonstrate that EGO-Prompt consistently outperforms previous methods, achieving an average F1 improvement of 7.32%–12.61% over the strongest baseline. Moreover, EGO-Prompt enhances the performance of lightweight models such as GPT-4o mini, surpassing reasoning models like o4-mini and o1, despite their inference costs being 6 to 140 times higher.

## 2 Related Works

**Generalized Multi-step Reasoning.** Currently, enhancing the multi-step reasoning capabilities of LLMs is primarily achieved through two turning-free approaches [34]: 1) In-Context Learning (ICL) [35–37] and Chain-of-Thought (CoT) [14, 15, 38], 2) Prompt Optimization [39, 40]. The open-ended nature of the reasoning space in ICL makes identifying optimal demonstrations and paths challenging [38, 41]. Prompt optimization aims to find the best prompt to guide LLMs towards better reasoning, with Automatic Prompt Optimization (APO) automating this process through candidate generation, evaluation, and filtering [13, 42]. A recent APO framework, TextGrad [16], draws inspiration from deep learning optimization methods (e.g., PyTorch [43]). However, it is highly prone to overfitting the training set after several iterations, where the adjusted prompt focuses on case-by-case details rather than general feature distribution.

Recent works have also explored using Reinforcement Learning (RL) to edit and optimize prompts for LLM reasoning [44–52]. TEMPERA [46] employs an attention-based policy model to edit prompts for LLMs and uses the logits difference between the LLM's predictions and the ground truth [45] as the reward function to fine-tune the policy. This approach effectively improves performance by producing

higher-quality prompts. However, such methods rely on access to the model's logits during training. Recent studies have extended this RL framework to black-box LLMs by adopting logit-free reward designs [50, 49]. While these RL methods can be more stable after careful fine-tuning compared to APO methods (thanks to their numerical optimization framework) they still require fine-tuning a policy model. Importantly, the policy model must be reasonably strong, ideally not much weaker than the target LLM (typically 7B parameters or larger), which imposes substantial demands on fine-tuning expertise and computational resources. In our tasks, the motivation is to leverage available LLMs for better adaptation to domain-specific applications (e.g., public health, transportation), where end users are often domain experts with limited access to large-scale computation.

Table 1: Taxonomy of related literature. The ✓ represents the technique is used or implemented by the method. The △ means the method needs manual designed prompts as seeds to start the optimization process.

| | Prompt Engineering | | Domain Knowledge Database | | Graph-Enhanced Reasoning | |
|---|---|---|---|---|---|---|
| | Manual Optimization | Automated Optimization | Knowledge Reference | Active Knowledge | Reasoning Guidance | Graph Correction |
| RAG [27] | ✓ | | ✓ | | | |
| ICL [35] | ✓ | | | | | |
| Zero-Shot-CoT [14] | ✓ | | | | | |
| APE [13] | △ | ✓ | | | | |
| ProTeGi [42] | △ | ✓ | | | | |
| CoK [31] | ✓ | | ✓ | | | |
| Li et al. [24] | ✓ | | ✓ | | | |
| RoG [25] | ✓ | | ✓ | | ✓ | |
| PHP [37] | ✓ | | | | | |
| $G^2$-Reasoner[26] | ✓ | | ✓ | | ✓ | |
| TextGrad [16] | △ | ✓ | | | | |
| Luo et al. [53] | ✓ | ✓ | ✓ | | ✓ | |
| **EGO-Prompt** | △ | ✓ | ✓ | ✓ | ✓ | ✓ |

**Domain Adaptive Reasoning.** LLMs often need external knowledge for effective reasoning in applicable or evolving domains (e.g., robotics [10], public health modeling [1], urban planning [54, 55], traffic safety [4, 56], autonomous driving [57, 58]). Retrieval-Augmented Generation (RAG) [27] is one of the solutions by retrieving relevant information from a corpus based on the input query. **However, suitable text corpora are not always available, and simply incorporating retrieved text does not guarantee improved reasoning processes** [59]. To address this, structured knowledge representations like **Knowledge Graphs and Causal Graphs** offer promising alternatives. Knowledge Graph can explicitly represent entities and relations, enabling more structured reasoning guidance compared to raw text retrieval. Existing methods focus on retrieve related paths from knowledge graph database to guide LLMs' reasoning [24, 31]. Causal Graph, in particular, offer detailed causal information distinct from general KGs [53, 60]. Most existing mentioned graph integration methods are static, utilizing pre-defined graph database, which limits the generalization capabilities of LLMs and may introduce biases originating from domain-specific graphs (see Table 1). Therefore, the primary objective of this research is to develop an evolutionary information integration approach that organically merges structural graph priors with the flexibility of textual information, thereby enabling an optimal reasoning process for real-world domain applications.

## 3  Prompt Optimization through Textual Gradients

Textual gradients is one type of automatic prompt optimization method that leverages the natural language feedback generated by LLMs to iteratively refine and enhance various components of AI systems [42, 16]. The core idea is to emulate the forward-backward learning paradigm in deep learning frameworks such as PyTorch [43], enabling the system to update prompts through a loop of evaluation and revision. The entire process can be divided into a textual forward and a textual backward phase [61]:

**Textual Forward.** For a classification task, given a system prompt $\mathcal{P}_{\text{sys}}$, input data $x_i$, and its corresponding label $y_i$, the forward model $\mathcal{M}_F$ generates a prediction $\hat{y}_i = \mathcal{M}_F(x_i; \mathcal{P}_{\text{sys}})$.

**Textual Backward.** Distinct from traditional numerical learning frameworks, the textual gradients method [42, 16] leverages a text-based loss function $\mathcal{L}$ to evaluate the alignment between the prediction $\hat{y}_i$ and the ground truth $y_i$. For example:

$$\mathcal{L}(\hat{y}_i, y_i) = \begin{cases} \texttt{"Prediction matches the ground truth.",} & \text{if } \hat{y}_i = y_i \\ \texttt{"Prediction does not match.",} & \text{otherwise} \end{cases} \quad (2)$$

In conventional deep learning frameworks such as PyTorch [43], gradients are computed numerically and used to update parameters via gradient descent. In contrast, textual gradients emulate this process by using LLMs to generate natural language feedback that guides prompt revision such as `The prompt can be improved by [strategies]`. The textual gradient of $\mathcal{P}_{\text{sys}}$ with respect to the loss is defined as:

$$\nabla_{\mathcal{P}_{\text{sys}}} \mathcal{L}_i = \frac{\partial \mathcal{L}_i}{\partial \mathcal{P}_{\text{sys}}} = \frac{\partial \mathcal{L}_i}{\partial \hat{y}_i} \cdot \frac{\partial \hat{y}_i}{\partial \mathcal{P}_{\text{sys}}} \quad \text{(Chain Rule)}$$

$$= \mathcal{M}_B\left(\mathcal{P}_{\text{sys}}, \hat{y}_i, \frac{\partial \mathcal{L}_i}{\partial \hat{y}_i}\right) \quad \text{(Implementation)},$$

(3)

where $\mathcal{M}_B$ denotes the **textual backward engine**, typically stronger than the forward model $\mathcal{M}_F$. The system prompt for the $\mathcal{M}_B$ is omitted here. The quantity $\frac{\partial \mathcal{L}_i}{\partial \hat{y}_i} = \mathcal{M}_B(\hat{y}_i, \mathcal{L}_i)$ represents the feedback generated by the backward engine, indicating the improvement direction for the predictions. This gradient follows from the chain rule and can be accumulated across iterations by concatenating past gradients [16]. The updated system prompt $\mathcal{P}'_{\text{sys}}$ is then obtained by applying the textual gradient:

$$\mathcal{P}'_{\text{sys}} = \mathcal{M}_B(\mathcal{P}_{\text{sys}}, \nabla_{\mathcal{P}_{\text{sys}}} \mathcal{L}_i) \tag{4}$$

## 4 Domain-Specific Reasoning with Expert Knowledge Guidance

### 4.1 Human-guided Graph Initialization

**Textualization of Raw Data.** Domain-specific tasks often involve knowledge and information in heterogeneous formats, such as numerical, textual and tabular data. To enable LLMs to process the structured data, a common first step is to construct manually designed templates that convert relevant information into textual prompts [1, 4]. Prompt 4.1 shows an example of partial prompt used in crash prediction task. The full prompt can be found in the Appendix 8.7.

---

**Prompt 4.1: Organized Prompt Example of Crash Prediction Task**

Predict the crash severity based on the crash event details [. . . ]
**[Time]** The crash occurred on April 29, 2022 at hour 16.
**[Dynamic Conditions]** The light condition is Daylight. The weather condition is Clear. [. . . ]

---

**Graph Establishment.** To better leveraging the reasoning process of LLMs with causal information, we propose the **Semantic Causal Graph (SCG)** as a Directed Acyclic Graph (DAG), where nodes represent entities or events extracted from the organized prompt, and edges denote causally-related semantic relations inferred from expert knowledge. Since this graph is not used for strict causal inference, we do not require it to satisfy the causal Markov assumption or the faithfulness assumption [62]. Formally, SCG for a domain-specific task can be represented as $\mathcal{G} = \{(n_i, r_{ij}, n_j) \mid n_i, n_j \in \mathcal{N}, r_{ij} \in \mathcal{R}\}$, where $\mathcal{N}$ is the set of semantic nodes and $\mathcal{R}$ is the set of semantic causal relations. Each node $n_i \in \mathcal{N}$ corresponds to a information block extracted from the organized prompt. Each edge $r_{ij} \in \mathcal{R}$ denotes a directed causal relation from $n_i$ to $n_j$, capturing the semantic causal link between them as expressed in natural language. In EGO-Prompt, the initial construction of the SCG relies on expert knowledge or external data analysis.

The biggest difference between the proposed work and most existing works is that the original input of the SCG does not need to be perfectly accurate or complete, given the evolutionary process of graph growth. Automatic correction will be involved and discussed in the Section 6.2. Below is an example SCG prompt for the crash prediction task and a full SCG is shown in the Appendix 8.8.

---

**Prompt 4.2: SCG Example for Crash Prediction Task**

CAUSAL STATEMENT 1: **[Person Status]** affects **[Severity]**.
The driver's Blood Alcohol Content (BAC) significantly increases the probability of fatal crashes. [. . . ]

---

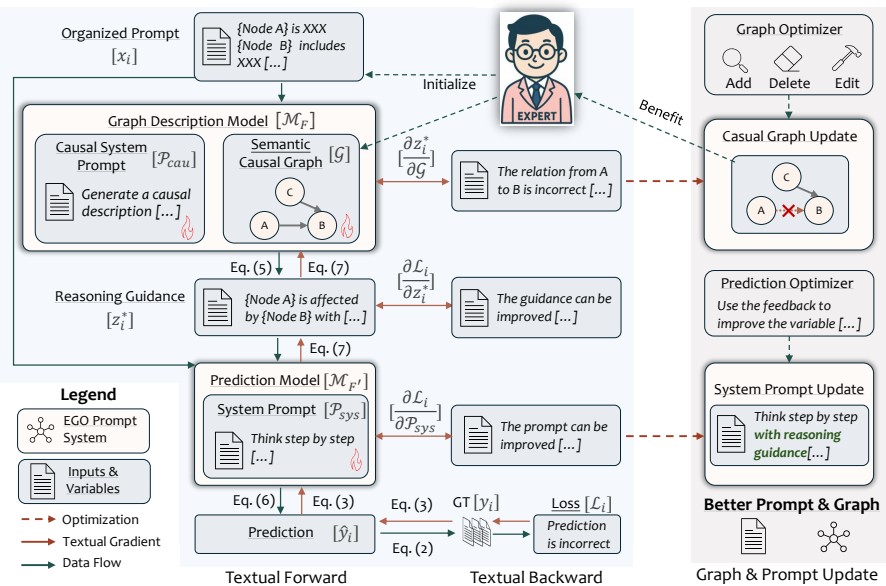

Figure 2: The optimization process of EGO-Prompt. The graph model generates reasoning guidance from the SCG and input prompt, which the prediction model uses to produce an output. Textual gradients are used to update the system prompt and refine the SCG through targeted operations.

## 4.2 Reasoning with Instance-specific Guidance

Given the expert-constructed SCG $\mathcal{G}$, our goal is to maximize the likelihood of the ground-truth label $y$ conditioned on the input $x$ and $\mathcal{G}$, i.e. $p(y \mid x, \mathcal{G})$. Because $\mathcal{G}$ is a global description that (i) always contains information irrelevant to the current case and (ii) can be partially missing for a particular instance (e.g. the BAC field may be None in Prompt 4.2), feeding it directly to the predictor is sub-optimal. Instead, as shown in Eq. (1), we first distill instance-specific reasoning guidance, denoted $z^*(x, \mathcal{G})$, from the graph $\mathcal{G}$ based on the input $x$. This guidance is then used for predicting the final results. As shown in Fig. 2, a graph description model $\mathcal{M}_F$ produces the reasoning guidance $z_i^*$ for each input $x_i$:

$$z_i^* = \mathcal{M}_F\big(x_i; \mathcal{P}_{\text{cau}}, \mathcal{G}\big), \tag{5}$$

where $\mathcal{P}_{\text{cau}}$ is a causal-system prompt steering $\mathcal{M}_F$ to ground its deterministic generation process with $x_i$ and $\mathcal{G}$. The predictor $\mathcal{M}_{F'}$ then reasons jointly over $x_i$ and $z_i^*$:

$$\hat{y}_i = \mathcal{M}_{F'}\big(x_i, z_i^*; \mathcal{P}_{\text{sys}}\big), \tag{6}$$

with $\mathcal{P}_{\text{sys}}$ instructing $\mathcal{M}_{F'}$ to integrate the original description and the generated reasoning guidance. This two-stage factorization filters out extraneous or missing details in $\mathcal{G}$ while preserving its causal structure, yielding cleaner and more informative context for the final prediction (see Section 6.1).

## 4.3 Optimizing SCG and Reasoning Process Through Textual Gradients

**Optimization through Textual Gradients.** Ideally, a well-constructed SCG and an effective system prompt can substantially enhance the model's reasoning process and predictive performance. However, the performance may vary due to two issues: (1) the SCG is incomplete or flawed, and (2) the system prompt fails to guide the model effectively. To address this, we innovated textual gradients method with graph priors [16, 42] to update flawed SCGs and optimize the system prompt, thereby enabling more effective SCG-based reasoning. We define the loss function $\mathcal{L}_i = \mathcal{L}(\hat{y}_i, y_i)$ similar to Eq. (2) to evaluate the difference between the final prediction $\hat{y}_i$ and the ground truth $y_i$. To improve the system prompt, we compute its textual gradients $\nabla_{\mathcal{P}_{\text{sys}}} \mathcal{L}_i$ using the chain rule in Eq. (3), which estimates how changes in the system prompt $\mathcal{P}_{\text{sys}}$ influence the loss through the model's output. The textual gradients for the SCG $\mathcal{G}$ and the causal system prompt $\mathcal{P}_{\text{cau}}$ can be formulated as:

$$\nabla_{\mathcal{G}} \mathcal{L}_i = \nabla_{z_i^*} \mathcal{L}_i \cdot \frac{\partial z_i^*}{\partial \mathcal{G}}, \quad \nabla_{\mathcal{P}_{\text{cau}}} \mathcal{L}_i = \nabla_{z_i^*} \mathcal{L}_i \cdot \frac{\partial z_i^*}{\partial \mathcal{P}_{\text{cau}}} \tag{7}$$

where $\nabla_{z_i^*} \mathcal{L}_i = \frac{\partial \mathcal{L}_i}{\partial \hat{y}_i} \cdot \frac{\partial \hat{y}_i}{\partial z_i^*}$ is the textual gradients for the reasoning guidance $z_i^*$.

With textual gradients, the system prompt, SCG, and causal system prompt can be updated based on Eq. (4). As shown in Figure 2, SCG refinement is constrained to three operations: (1) **Add** a node from the candidate set $\mathcal{N}$ with its causal description and links; (2) **Delete** a node and its associated descriptions; and (3) **Edit** existing descriptions which is incorrect or unnecessary.

**Iterative Optimization.** The goal of the proposed optimization is to identify the optimal system prompt $\mathcal{P}_{sys}$, SCG $\mathcal{G}$, and causal system prompt $\mathcal{P}_{cau}$. These components are co-optimized by two optimizers in our framework (see Figure 2), each governed by distinct update rules. Performing all updates with a single LLM in a single iteration may lead to suboptimal performance due to conflicting update signals. Therefore, we adopt an iterative optimiza-

---

**Algorithm 1** EGO-Prompt Iterative Optimization

**Require:** $P_{sys}^0, P_{cau}^0, \mathcal{G}^0$, dataset $\mathcal{D} = \{(x_i, y_i)\}_{i=1}^N$, steps $T$
1: $P_{sys}, P_{cau}, \mathcal{G} \leftarrow P_{sys}^0, P_{cau}^0, \mathcal{G}^0$
2: Test on validation set and get F1 $f \leftarrow F_1(P_{sys}, P_{cau}, \mathcal{G}; \mathcal{D})$
3: **for** $t = 1$ **to** $T$ **do**
4:     Sample $(x_i, y_i) \sim \mathcal{D}$; derive reasoning guidance $z_i^*$ (Eq. (5))
5:     $\hat{y}_i \leftarrow$ Eq. (6);    $\mathcal{L}_i \leftarrow$ Eq. (2)
6:     $(\nabla_{sys}, \nabla_{cau}, \nabla_{SCG}) \leftarrow$ Eqs. (3) and (7)
    **Stage 1: Update system prompt**
7:     $P_{sys}' \leftarrow$ Apply$(P_{sys}, \nabla_{sys})$
8:     $f' \leftarrow F_1(P_{sys}', P_{cau}, \mathcal{G})$
9:     **if** $f' > f$ **then** $P_{sys}, f \leftarrow P_{sys}', f'$
10:     **end if**
    **Stage 2: Update SCG & causal prompt**
11:     $P_{cau}' \leftarrow$ Apply$(P_{cau}, \nabla_{cau})$; $\mathcal{G}' \leftarrow$ Apply$(\mathcal{G}, \nabla_{SCG})$
12:     $f' \leftarrow F_1(P_{sys}, P_{cau}', \mathcal{G}')$
13:     **if** $f' > f$ **then** $(P_{cau}, \mathcal{G}, f) \leftarrow (P_{cau}', \mathcal{G}', f')$
14:     **end if**
15: **end for**
16: **return** $(P_{sys}, \mathcal{G}, P_{cau})$

---

tion strategy that updates the SCG and prompt components separately. One component is updated only when it yields an individual performance improvement while the other component is fixed. We perform a fixed number of optimization steps per task. The full optimization procedures are shown in Algorithm 1.

## 5 Experiments on Three Domain Tasks

**Experimental Objectives for Testing the Proposed EGO Prompt.** We evaluate the proposed EGO Prompt based on three main criteria: 1) Performance Improvement: compare the performance of EGO Prompt with existing prompt-optimization frameworks to determine whether it yields better results. 2) Generalization: assess how well EGO Prompt generalizes across a wide range of real-world tasks on diverse LLM model zoos [63, 64]. 3) Efficiency and Cost-Effectiveness: examine whether a smaller-parameter, lower-reasoning-capacity model (e.g., GPT-4o mini) can achieve comparable performance to a larger model (e.g., o4-mini or o1) after optimization using EGO Prompt. Measure the resulting cost savings relative to the performance level of the larger model.

### 5.1 Datasets for Public Health, Transportation, and Human Behavior Modeling

To assess the generalization and cross-domain adaptivity of EGO Prompt, we evaluate its performance using three real-world applications with publicly available datasets: Pandemic [1], TrafficSafe [4], and Swissmetro [65]. These datasets span the domains of public health, traffic safety, and human behavior, respectively. LLMs have been extensively utilized in these domains due to their capability to interpret complex textual inputs (e.g., crash reports) and generalize knowledge across diverse scenarios (e.g., from COVID-19 to future pandemics). Specifically, as shown in table in Table 2: **1) Public Health - Pandemic Hospitalization Dataset** [1, 66] consists of textual descriptions for forecasting pandemic hospitalization trends. It is constructed from CDC COVID-19 reports [67], reformulated into human-designed prompts. Each instance includes demographic information, COVID-19 case counts, vaccination rates, and hospitalization patterns for individual states, with the target variable being the hospitalization trend for the subsequent week. **2) Crash Modeling for Transportation – TrafficSafe Dataset** [4, 68–71] comprises structured textual records derived from real-world crash reports across the United States [72]. Each entry provides details about crash time, location, weather conditions, road surface conditions, vehicle maneuvers, and driver behaviors, paired with the corresponding crash severity outcomes. **3) Human Behavior Modeling – Travel Mode Choice Dataset** [65, 73] originates from a stated-preference survey conducted in Switzerland. It presents respondents with hypothetical travel scenarios, prompting them to choose

between Swissmetro, traditional trains, and cars based on various factors including travel time, cost, and comfort.

Table 2: Overview of the Datasets Employed in This Study.

| Dataset | Domain | Prediction Targets | Domain Task labels | Dataset Size |
|---|---|---|---|---|
| Pandemic [1] | Public Health | Pandemic Trends | substantial decreasing, moderate decreasing, stable, moderate increasing, substantial increasing | 5,200 |
| TrafficSafe [4] | Traffic Safety | Traffic Crash Severity | no apparent injury, minor injury, serious injury, fatal | 16,188 |
| Swissmetro [65] | Human Behavior | Travel Mode Choice | swissmetro, car, train | 10,728 |

## 5.2 Settings

**Experimental Settings.** Following prior work on automatic prompt optimization [16, 42], we randomly sample a balanced subset from each dataset: 100 instances for validation, 100 for testing, and the rest for training. Due to the inherent stochasticity of LLM API calls (see Section 8.5), we repeat each experiment three times and report the best result to represent the achievable upper-bound performance under identical settings (same random seed). We report weighted F1 and accuracy, using a batch size 3 to perform 6 to 12 optimization steps per task.

**Models.** We use `gpt-4o-2024-08-06` [63] as the backward engine $\mathcal{M}_B$ across all experiments, as it is considered more powerful than the forward engine [74]. For the forward engine $\mathcal{M}_F$, we evaluate several mainstream commercial models [63, 64], including `gpt-4o-mini-2024-07-18`, `gpt-4.1-mini-2025-04-14`, `gemini-2.0-flash`, and `gemini-2.5-flash-preview-04-17`. The graph description model $\mathcal{M}_{F'}$ is always set to be the same as $\mathcal{M}_F$.

**Baseline Methods.** We compare our approach against two APO methods: 1) **ProTeGi** [42]. Pro-TeGi is the first framework to use natural language "gradients" to generate feedback for prompt optimization, combining beam search with bandit-based selection to identify the optimal prompts. 2) **TextGrad** [16]. This method is introduced in Section 3. We further incorporate three prompt engineering methods: 3) **Zero-Shot-CoT** approach [14], which enables step-by-step reasoning without in-context examples; 4) **PHP** [37], which progressively integrates historical model predictions as hints appended to the prompt to guide reasoning; and 5) **Auto-CoT** [75], which is an automatic prompting method that generates diverse CoT demonstrations using LLMs. RAG-based methods are excluded due to the lack of accessible domain-specific graph databases for our datasets. We also include the 6) **Expert Organized Prompt**, which uses the original input (specifically, the initial $x_i$ and $\mathcal{P}_{\text{sys}}$) to make direct predictions.

## 5.3 Results Comparison and Summary

**Comparison with Cutting-edge Baselines.** Table 3 shows the performance of EGO-Prompt compared to baselines on the Pandemic, TrafficSafe, and Travel Mode Choice datasets. EGO-Prompt consistently outperforms all methods, achieving average F1 gains of 7.32% with GPT-4o mini, 12.61% with Gemini 2.5 Flash, and 9.07% with GPT-5-mini over the best baseline. In contrast, some methods such as TextGrad show limited or even negative improvement in certain cases (e.g., Pandemic with GPT-4o mini, F1 score only improve from 0.347 to 0.359). This is primarily attributed to the tendency of these methods to overfit the validation set (e.g., potentially generating overly long prompts with TextGrad), leading to a performance decline on the test set.

**Generalization.** EGO-Prompt also brings consistent improvements across different backbone models. As shown in Figure 3, it enhances performance on all three tasks when using GPT-4o mini, GPT-4.1 mini, GPT-5o mini, Gemini 2.0 Flash, and Gemini 2.5 Flash as the forward engine. Specifically, EGO-Prompt achieves relative F1 score gains ranging from 13.3% to 37.0% compared to the organized prompt baseline, showing robust generalization capability across both models and domains. See Appendix 8.2 for more results for commercial and open-source models.

**Efficiency and Cost-Effectiveness.** EGO-Prompt enables smaller LLMs to rival or exceed the accuracy of larger, costlier models. As illustrated in Figure 3, EGO-Prompt effectively boosts the average performance of compact models, matching the effectiveness of the reasoning model o4-mini. Moreover, as shown in Figure 4, the mean inference cost per 100 samples using EGO-Prompt is only $0.057, substantially lower than that of o4-mini ($0.33) and other reasoning models.

Table 3: Performance comparison with baselines.

| Forward Engine | Method | Venue/Journal | Pandemic | | TrafficSafe | | Mode Choice | | Mean F1 | |
|---|---|---|---|---|---|---|---|---|---|---|
| | | | Acc | F1 | Acc | F1 | Acc | F1 | Value | Imp. |
| GPT-4o mini | Organized Prompt | — | 0.360 | 0.347 | 0.300 | 0.232 | 0.459 | 0.406 | 0.328 | — |
| | Zero-Shot-CoT [14] | NeurIPS'22 | 0.370 | 0.361 | 0.280 | 0.221 | 0.494 | 0.435 | 0.339 | 3.2% |
| | ProTeGi [42] | EMNLP'23 | 0.370 | 0.361 | 0.370 | 0.304 | **0.529** | 0.481 | 0.382 | 16.3% |
| | Auto-CoT [75] | ICLR'23 | 0.380 | 0.352 | 0.320 | 0.220 | 0.447 | 0.462 | 0.345 | 5.0% |
| | PHP [37] | ICLR'24 | 0.330 | 0.327 | 0.320 | 0.268 | 0.376 | 0.370 | 0.322 | -2.0% |
| | TextGrad [16] | Nature'25 | 0.380 | 0.359 | 0.300 | 0.243 | 0.506 | 0.432 | 0.345 | 5.0% |
| | **EGO-Prompt** | This work | **0.410** | **0.399** | **0.380** | **0.333** | 0.506 | **0.498** | **0.410** | **24.9%** [2] |
| Gemini 2.5 Flash | Organized Prompt | — | 0.470 | 0.470 | 0.340 | 0.319 | 0.459 | 0.392 | 0.394 | — |
| | Zero-Shot-CoT [14] | NeurIPS'22 | 0.490 | 0.490 | 0.400 | 0.390 | 0.482 | 0.398 | 0.426 | 8.1% |
| | ProTeGi [42] | EMNLP'23 | 0.470 | 0.482 | 0.420 | 0.400 | 0.494 | 0.425 | 0.435 | 10.6% |
| | Auto-CoT [75] | ICLR'23 | 0.380 | 0.352 | 0.340 | 0.230 | 0.447 | 0.462 | 0.348 | 6.0% |
| | PHP [37] | ICLR'24 | 0.520 | 0.515 | 0.360 | 0.334 | 0.494 | 0.436 | 0.428 | 8.6% |
| | TextGrad [16] | Nature'25 | 0.470 | 0.483 | 0.380 | 0.374 | 0.494 | 0.428 | 0.428 | 8.6% |
| | **EGO-Prompt** | This work | **0.540** | **0.546** | **0.430** | **0.428** | 0.518 | **0.499** | **0.491** | **24.6%** |
| GPT-5-mini | Organized Prompt | — | 0.420 | 0.387 | 0.330 | 0.265 | 0.435 | 0.435 | 0.362 | — |
| | Zero-Shot-CoT [14] | NeurIPS'22 | 0.430 | 0.415 | 0.330 | 0.281 | 0.424 | 0.428 | 0.375 | 3.4% |
| | ProTeGi [42] | EMNLP'23 | 0.420 | 0.413 | 0.320 | 0.267 | 0.471 | 0.454 | 0.378 | 4.4% |
| | Auto-CoT [75] | ICLR'23 | 0.410 | 0.408 | 0.310 | 0.280 | 0.506 | 0.470 | 0.386 | 6.6% |
| | PHP [37] | ICLR'24 | 0.430 | 0.397 | 0.320 | 0.240 | 0.435 | 0.431 | 0.356 | -1.8% |
| | TextGrad [16] | Nature'25 | 0.430 | 0.415 | 0.250 | 0.230 | 0.494 | 0.485 | 0.377 | 4.0% |
| | **EGO-Prompt** | This work | **0.460** | **0.448** | **0.340** | **0.305** | **0.529** | **0.511** | **0.421** | **16.3%** |

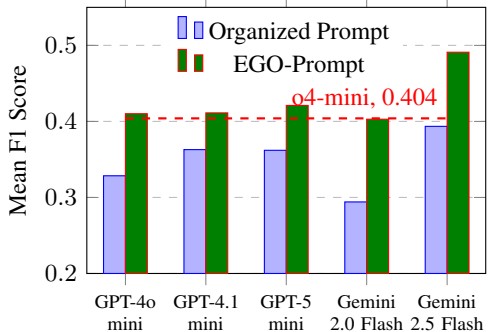

Figure 3: Mean F1 Score Across 3 Datasets.

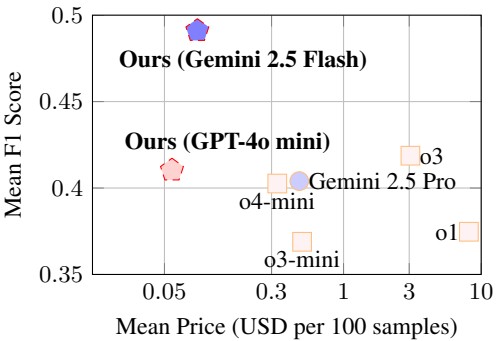

Figure 4: Mean F1 Score and Mean Price with Reasoning Models.

# 6 Analysis and Insights

## 6.1 Ablation Study

**Instance-specific Reasoning Guidance.** The instance-specific reasoning guidance mechanism in Eq. (1) plays a critical role in enhancing optimization effectiveness. As shown in Table 4, removing the decomposition process in Eq. (1) and using a single model leads to noticeably worse performance. Specifically, the F1 scores across the three tasks drop from 0.399, 0.333, and 0.498 to 0.397, 0.247, and 0.445, respectively.

Table 4: Ablation Study of Instance-specific Reasoning Guidance. GPT-4o mini is used for all the ablation study and analysis experiments.

| Base Model | Pandemic | | TrafficSafe | | Swissmetro | |
|---|---|---|---|---|---|---|
| | Acc | F1 | Acc | F1 | Acc | F1 |
| Single Model | 0.390 | 0.397 | 0.340 | 0.247 | 0.482 | 0.445 |
| **EGO-Prompt** | **0.410** | **0.399** | **0.380** | **0.333** | **0.506** | **0.498** |

**Model Components.** We conduct an ablation study to evaluate the contribution of the graph description model (with variables $\mathcal{G}$ and $\mathcal{P}_{\text{cau}}$) and the prediction model (with variable $\mathcal{P}_{\text{sys}}$) to the overall performance. As shown in Table 5, our full framework, where both components are used and updated during optimization, achieves the best performance across all three tasks. Fixing either the graph description model or the prediction model leads to noticeable performance drops. Furthermore, removing the graph description model entirely (i.e., only using the prediction model, similar to the TextGrad method [16]) results in consistently lower F1 scores.

---

[2] See Appendix 8.5 for the distribution.

Table 5: Ablation Study of Model Components. A checkmark (✓) indicates the component is used and updated during optimization, while a triangle (△) indicates it is used but kept fixed.

| $\mathcal{G}$ & $\mathcal{P}_{cau}$ | $\mathcal{P}_{sys}$ | Pandemic | | TrafficSafe | | Swissmetro | |
|---|---|---|---|---|---|---|---|
| | | Acc | F1 | Acc | F1 | Acc | F1 |
| | ✓ | 0.380 | 0.359 | 0.300 | 0.243 | 0.506 | 0.432 |
| △ | ✓ | 0.370 | 0.337 | 0.270 | 0.246 | 0.471 | 0.418 |
| ✓ | △ | 0.370 | 0.354 | 0.330 | 0.257 | 0.471 | 0.418 |
| ✓ | ✓ | **0.410** | **0.399** | **0.380** | **0.333** | **0.506** | **0.498** |

**Iterative Optimization.** We conduct an ablation study to assess the effectiveness of our optimization strategy by comparing three variants: (1) removing the optimization process entirely (W/O OPT.), (2) disabling the iterative optimization process (W/O ITERATIVE OPT.), and (3) using the full strategy in EGO-Prompt. As shown in Table 6, the absence of optimization results in reduced performance, with F1 scores dropping to 0.359, 0.293, and 0.434 on three tasks, respectively. When iterative refinement is removed, the F1 scores are moderately improved (0.369, 0.272, and 0.447) but still lag behind the full model.

Table 6: Ablation Study of Iterative Optimization.

| Base Model | Pandemic | | TrafficSafe | | Swissmetro | |
|---|---|---|---|---|---|---|
| | Acc | F1 | Acc | F1 | Acc | F1 |
| W/O OPT. | 0.360 | 0.359 | 0.340 | 0.293 | 0.471 | 0.434 |
| W/O ITERATIVE OPT. | **0.410** | 0.369 | 0.310 | 0.272 | 0.447 | 0.447 |
| **EGO-Prompt** | **0.410** | **0.399** | **0.380** | **0.333** | **0.506** | **0.498** |

## 6.2 Automatic SCG Correction

EGO-Prompt automatically refines a human-designed SCG during iterative optimization by restricting the graph description updates to three operations: addition, deletion, and modification of causal description. These targeted updates correct biases or flaws in the expert-constructed SCG, guided by ground-truth data. Figure 5 illustrates this correction process on the Pandemic dataset, where EGO-Prompt identifies and incorporates missing connections (e.g., from `Healthcare System Condition` to `Hospitalization per 100k`) and removes weak or incorrect ones (e.g., from `Demographic Information` to `Restriction Policy Response`). The system prompt is also co-optimized throughout

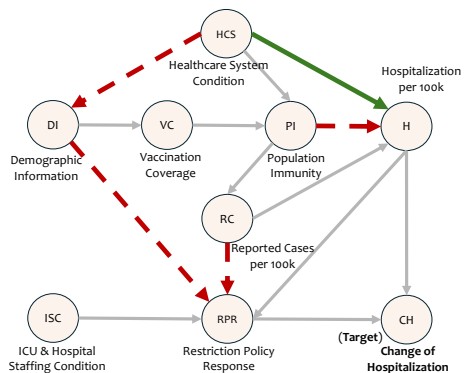

Figure 5: Automatic SCG Correction for the Pandemic Dataset. Green line denotes newly added relations; red dash denotes deleted relation.

this process. Additional corrected SCGs and adjusted system prompts for other datasets are provided in Appendix 8.9.

## 7 Conclusion

We propose a novel method called EGO-Prompt, which integrates expert knowledge to enhance the adaptivity and interpretability of LLMs for domain-specific tasks. By decomposing graph-based reasoning into two stages: generation of reasoning guidance and model reasoning conditioned on that guidance, EGO-Prompt not only outperforms existing prompt optimization baselines, but also enables automatic correction of the SCG and refinement of the reasoning process. In future work, EGO-Prompt holds promise not only for a broader range of domain-specific applications but also in other emerging directions such as Dynamic RAG (real-time updates to the graph database), domain-specific knowledge graph construction, and causal discovery. One noticeable limitation of EGO-Promptis that it requires additional computational resources for causal-guided textual gradient, especially when scaling to larger sample sets. Another limitation is the results can be unstable due to API variability and the inherent sensitivity of the Textual Gradients method (see Appendix 8.5 and 8.6 for details).

## Acknowledgments

Yang Zhao acknowledge the fellowship from JHU + Amazon Initiative for Interactive AI (AI2AI).

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

# 8 Technical Appendices and Supplementary Material

## 8.1 Generation of Reasoning Guidance

**Derivation.** Let $x$ denote the structured prompt for the domain-specific task, $\mathcal{G}$ the semantic causal graph (SCG), and $z$ the reasoning guidance distilled from $(x, \mathcal{G})$. Starting from the product rule,

$$p(y \mid x, \mathcal{G}) = \sum_z p(y, z \mid x, \mathcal{G}) \tag{8}$$

$$= \sum_z p(y \mid x, \mathcal{G}, z)\, p(z \mid x, \mathcal{G}). \tag{9}$$

**(A1) Conditional independence.** Because the SCG $\mathcal{G}$ is identical for every data point, once $(x, z)$ are given, the remaining information in $\mathcal{G}$ is irrelevant to $y$:

$$p(y \mid x, \mathcal{G}, z) = p(y \mid x, z).$$

Applying (A1) to (9) yields

$$p(y \mid x, \mathcal{G}) = \sum_z p(y \mid x, z)\, p(z \mid x, \mathcal{G}). \tag{10}$$

**(A2) Deterministic guidance.** To guarantee that the graph description model $\mathcal{M}_{F'}$ yields a single, repeatable guidance for each $(x, \mathcal{G})$, we impose the following output constraint in the optimizer:

---
**Prompt 8.1: Output Constraint for the Graph-Description Model**

**Format**
```
<Causal Description>
```
Provide a numbered list of causal statements grounded in the supplied causal relations and crash details. Each statement must explicitly articulate the causal mechanism whenever it is available.
```
</Causal Description>
```

---

Because every instance is processed with the same causal system prompt $\mathcal{P}_{\text{cau}}$ and this constraint, the model behaves almost deterministically, inducing the mapping

$$z^*(x, \mathcal{G}) \;=\; \mathcal{M}_{F'}(x, \mathcal{G}; \mathcal{P}_{\text{cau}}).$$

so that the posterior over $z$ collapses to a Dirac delta, $p(z \mid x, \mathcal{G}) = \delta\big(z - z^*(x, \mathcal{G})\big)$. Substituting this into (10):

$$p(y \mid x, \mathcal{G}) = p\big(y \mid x, z^*(x, \mathcal{G})\big) \sum_z \delta\big(z - z^*(x, \mathcal{G})\big)$$

$$= p\big(y \mid x, z^*(x, \mathcal{G})\big). \tag{11}$$

Equation (11) shows that, under assumptions (A1)–(A2), the global graph $\mathcal{G}$ can be replaced by the sample-specific deterministic guidance $z^*(x, \mathcal{G})$ without losing any predictive information.

## 8.2 Performance Comparison of Open Source Models and Commercial Models

In addition to the commercial models GPT-4o-mini, GPT-4.1-mini, GPT-5-mini, Gemini 2.5 Flash, and Gemini 2.0 Flash, we extended our experiments to evaluate EGO-Prompt on open-source models, including Qwen3 (1.7B, 8B, 14B, 32B) [76], DeepSeek-V3 [77], and Llama-3.3 (8B, 70B), and Llama-4-Scout-17B [78]. Table 7 shows the experiment results. Several findings can be summarized:

- **EGO-Prompt is effective for open source models.** EGO-Prompt boosts the performance of open-source models by 49.9%, which is substantially higher than its improvement on commercial models (22.7%).

- **Commercial models achieve better and more stable performance.** Across both the initial organized prompt and the optimized settings, commercial models consistently outperform open-source models, achieving higher F1 scores and greater robustness across tasks compared with open-source models (0.427 vs. 0.351).

- **Model size substantially influences the performance ceiling.** Models with more parameters achieve better optimized performance, consistent with the scaling law. For example, the Qwen3 series improves from an F1 score of 0.265 for the 1.7B model to 0.390 for the 32B model. DeepSeek-V3, with 671B parameters, achieves the highest performance among the open-source models. These results suggest that our method can consistently enhance performance across models of varying sizes.

- **EGO-Prompt enhances reasoning in smaller models.** Models such as Qwen3-1.7B and Qwen3-8B fail to generate structured outputs under the initial organized prompt. However, with EGO-Prompt, they are able to produce more structured predictions and follow certain reasoning paths to make predictions, although their overall performance remains limited.

Table 7: Performance comparison of open source models and commercial models across three datasets.

| Type | Base Model | Pandemic | | TrafficSafe | | Mode Choice | | Mean F1 (↑%) |
|---|---|---|---|---|---|---|---|---|
| | | Acc | F1 | Acc | F1 | Acc | F1 | |
| Open Source | Qwen-1.7B | 0.120 | 0.158 | 0.110 | 0.120 | 0.047 | 0.084 | 0.121 |
| | Qwen-1.7B with EGO | 0.190 | 0.178 | 0.200 | 0.230 | 0.412 | 0.386 | **0.265 (↑119.5%)** |
| | Qwen-8B | 0.060 | 0.080 | 0.220 | 0.100 | 0.059 | 0.096 | 0.092 |
| | Qwen-8B with EGO | 0.140 | 0.199 | 0.290 | 0.202 | 0.459 | 0.427 | **0.276 (↑200.1%)** |
| | Qwen-14B | 0.360 | 0.299 | 0.330 | 0.227 | 0.424 | 0.309 | 0.279 |
| | Qwen-14B with EGO | 0.390 | 0.339 | 0.340 | 0.271 | 0.471 | 0.457 | **0.356 (↑27.7%)** |
| | Qwen-32B | 0.340 | 0.284 | 0.310 | 0.229 | 0.412 | 0.321 | 0.278 |
| | Qwen-32B with EGO | 0.430 | 0.405 | 0.350 | 0.285 | 0.494 | 0.481 | **0.390 (↑40.3%)** |
| | DeepSeek-V3 | 0.210 | 0.231 | 0.340 | 0.211 | 0.459 | 0.457 | 0.300 |
| | DeepSeek-V3 with EGO | 0.460 | 0.464 | 0.400 | 0.371 | 0.506 | 0.501 | **0.445 (↑48.7%)** |
| | LLaMA-3.3-8B | 0.350 | 0.274 | 0.310 | 0.210 | 0.310 | 0.210 | 0.231 |
| | LLaMA-3.3-8B with EGO | 0.370 | 0.326 | 0.330 | 0.289 | 0.377 | 0.335 | **0.316 (↑36.8%)** |
| | LLaMA-3.3-70B | 0.370 | 0.317 | 0.250 | 0.176 | 0.400 | 0.275 | 0.256 |
| | LLaMA-3.3-70B with EGO | 0.420 | 0.378 | 0.340 | 0.273 | 0.459 | 0.419 | **0.357 (↑39.5%)** |
| | Llama-4-Scout-17B | 0.360 | 0.307 | 0.260 | 0.237 | 0.294 | 0.287 | 0.277 |
| | Llama-4-Scout-17B with EGO | 0.420 | 0.376 | 0.320 | 0.268 | 0.482 | 0.459 | **0.368 (↑33.0%)** |
| | **Average** | 0.279 | 0.248 | 0.271 | 0.193 | 0.313 | 0.262 | 0.235 |
| | **Average with EGO** | 0.361 | 0.341 | 0.324 | 0.275 | 0.461 | 0.438 | **0.351 (↑49.9%)** |
| Commercial | GPT-4o-mini | 0.360 | 0.347 | 0.300 | 0.232 | 0.459 | 0.406 | 0.328 |
| | GPT-4o-mini with EGO | 0.410 | 0.399 | 0.380 | 0.333 | 0.506 | 0.498 | **0.410 (↑24.9%)** |
| | GPT-4.1-mini | 0.400 | 0.374 | 0.350 | 0.246 | 0.506 | 0.468 | 0.363 |
| | GPT-4.1-mini with EGO | 0.430 | 0.420 | 0.370 | 0.292 | 0.553 | 0.522 | **0.411 (↑13.2%)** |
| | GPT-5-mini | 0.420 | 0.387 | 0.330 | 0.265 | 0.435 | 0.435 | 0.362 |
| | GPT-5-mini with EGO | 0.460 | 0.448 | 0.340 | 0.305 | 0.529 | 0.511 | **0.421 (↑16.3%)** |
| | Gemini-2.5-Flash | 0.470 | 0.470 | 0.340 | 0.319 | 0.459 | 0.392 | 0.394 |
| | Gemini-2.5-Flash with EGO | 0.540 | 0.546 | 0.430 | 0.428 | 0.518 | 0.499 | **0.491 (↑24.6%)** |
| | Gemini-2.0-Flash | 0.300 | 0.268 | 0.330 | 0.264 | 0.259 | 0.351 | 0.294 |
| | Gemini-2.0-Flash with EGO | 0.430 | 0.412 | 0.400 | 0.327 | 0.506 | 0.470 | **0.403 (↑37.0%)** |
| | **Average** | 0.390 | 0.369 | 0.330 | 0.265 | 0.424 | 0.410 | 0.348 |
| | **Average with EGO** | 0.454 | 0.445 | 0.384 | 0.337 | 0.522 | 0.500 | **0.427 (↑22.7%)** |

### 8.3 Human-Expert Involvement Evaluation

To assess how human involvement and the completeness of the SCG affect the reasoning process, we evaluate our method under several SCG completeness settings (Table 8): 1) **Reversed SCG**. To test sensitivity to incorrect causal structures, we reversed all edges in the expert-designed SCGs used in the main experiments (see Figure 6). This allows us to compare performance differences between mostly correct and mostly incorrect SCGs. 2) **Empty SCG**. We removed the initial SCG and let the model construct it during optimization. This tests whether providing a good initial SCG improves performance and whether SCG can be constructed automatically. 3) **33% and 66% SCG**. These settings randomly retain only 33% or 66% of the original cauasl edges, reflecting different degrees of SCG completeness. The key observations are as follows:

- **Incorrect SCG degrades reasoning performance.** On the Pandemic dataset, the reversed SCG substantially degrades the performance (F1 from 0.359 to 0.303), suggesting that incorrect prior knowledge can misguide the optimization process.

- **Randomly removing causal edges can impair the model's reasoning capability.** The initial SCG represents a complete reasoning path. Although it may not be entirely correct, it maintains structural integrity and connected as a DAG. Random removal of edges disrupts this structure, leading to degraded performance. For example, the 33% SCG setting yields a lower mean F1 score than the Empty SCG, indicating that a partially broken causal graph may be worse than having no guidance at all.

- **Greater completeness in the SCG supports better reasoning.** The 66% SCG setting achieves performance close to that of the full SCG, suggesting that more complete causal guidance helps the model reason more effectively.

- **If a user is unsure about the completeness or correctness of the SCG, it is advisable to start with an empty SCG and iteratively refine it.** This allows gradual construction of a reliable causal graph without risking the adverse effects of incorrect or incomplete edges.

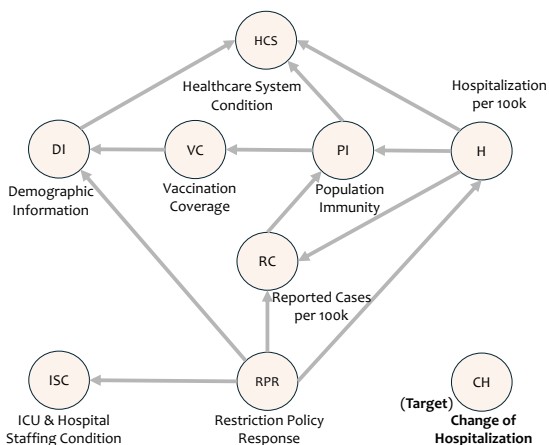

Figure 6: Reversed Pandemic SCG.

Table 8: Performance under different SCG settings. We use GPT-4o-mini as the forward engine.

| SCG Setting | Pandemic | | TrafficSafe | | Mode Choice | | Mean F1 |
|---|---|---|---|---|---|---|---|
| | Acc | F1 | Acc | F1 | Acc | F1 | |
| No SCG | 0.380 | 0.359 | 0.300 | 0.243 | 0.506 | 0.432 | 0.345 |
| Reversed | 0.310 | 0.303 | 0.290 | 0.260 | 0.482 | 0.457 | 0.350 |
| Empty | 0.370 | 0.372 | 0.330 | 0.317 | 0.506 | 0.470 | 0.394 |
| 33% | 0.390 | 0.389 | 0.300 | 0.270 | 0.459 | 0.462 | 0.378 |
| 66% | 0.400 | 0.387 | 0.310 | 0.314 | 0.506 | 0.493 | 0.402 |
| **Full** | 0.410 | 0.399 | 0.380 | 0.333 | 0.506 | 0.498 | **0.421** |

## 8.4 Analyzing the Trade-off Between Performance and Cost

As shown in Table 9, we evaluate the inference performance and cost of our model compared to mainstream reasoning models, including OpenAI models (`o3-mini-2025-01-31`, `o4-mini-2025-04-16`, `o1-2024-12-17`, `o3-2025-04-16`) and Google's `gemini-2.5-pro-preview-05-06`. Costs for OpenAI models were recorded based on actual API platform expenses, while costs for Google models were estimated from input and output token counts. Results show that our EGO-Prompt (GPT-4o mini) matches the performance of o4-mini at roughly one-sixth the cost, and outperforms o1 at over one-hundredth the cost.

Table 9: Comparison of model performance with their price. Price is measured in USD per 100 inferences.

| Model | Pandemic | | | TrafficSafe | | | Swissmetro | | |
| --- | --- | --- | --- | --- | --- | --- | --- | --- | --- |
| | Acc | F1 | Price | Acc | F1 | Price | Acc | F1 | Price |
| o3-mini | 0.390 | 0.383 | $0.50 | 0.330 | 0.282 | $0.57 | 0.494 | 0.442 | $0.45 |
| o4-mini | 0.430 | 0.418 | $0.33 | 0.380 | 0.334 | $0.26 | 0.494 | 0.456 | $0.29 |
| o1 | 0.410 | 0.409 | $10.19 | 0.310 | 0.219 | $7.18 | 0.541 | 0.496 | $7.04 |
| o3 | 0.450 | 0.425 | $3.06 | 0.410 | 0.385 | $2.12 | 0.506 | 0.444 | $3.32 |
| Gemini 2.5 pro | 0.400 | 0.399 | $0.79 | 0.410 | 0.411 | $0.45 | 0.471 | 0.402 | $0.19 |
| EGO-Prompt (GPT-4o mini) | 0.410 | 0.399 | $0.04 | 0.380 | 0.333 | $0.08 | 0.506 | 0.498 | $0.05 |
| EGO-Prompt (Gemini 2.5 Flash) | 0.540 | 0.546 | $0.06 | 0.430 | 0.428 | $0.07 | 0.518 | 0.499 | $0.13 |

As our model optimizes the reasoning process using ground truth data, the training incurs additional cost. The cost of our method throughout the optimization process is evaluated using GPT-4o-mini as the forward engine and GPT-4o as the backward engine. Each optimization step includes two forward passes through both the graph description model and the prediction model, two backward passes to iteratively update the SCG, system prompt, and causal system prompt, as well as two evaluations on the validation set and zero or one evaluation on the test set (see Algorithm 1). We recorded the actual costs using the OpenAI API platform during the optimization and reported the average cost over three steps for each task. As shown in Table 10, each optimization step for the three tasks costs approximately \$0.3–\$0.4. Given that each task typically requires 6 to 12 optimization steps, the total cost for the full optimization process ranges from approximately \$2 to \$5 per task. However, in real-world domain-specific tasks, such as traffic safety modeling with hundreds of thousands of crashes reported annually in a single state, the training cost becomes relatively negligible compared to the inference cost.

Table 10: Optimization cost per step for each dataset, measured in USD.

| Dataset | Opt. Cost per Step (USD) |
| --- | --- |
| Pandemic | 0.310 |
| TrafficSafe | 0.403 |
| Swissmetro | 0.313 |

## 8.5 Variability of the Performance

Due to the inherent non-deterministic nature of LLM outputs (unstable even when temperature=0) [79], as well as the instability of the Textual Gradient method, our results exhibit fluctuations within a certain range. Figure 7 presents the box plots of the performance of the organized prompt and EGO-Prompt across 10 independent runs on three tasks. We observe that our method may exhibit up to 20% variability in performance across the three tasks. In some cases, as only a very limited number of training examples are used in our method, the model may overfit to the validation set in certain training step, leading to degraded performance on the test set. Addressing this issue will be an important direction for future work.

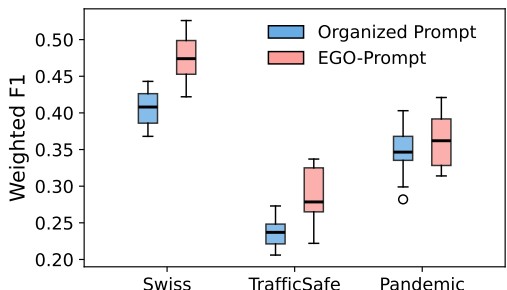

Figure 7: Performance of organized prompt and EGO-Prompt across 10 independent runs on three tasks. We use GPT-4o-mini as the forward engine.

## 8.6 Limitations

The primary limitation of our work lies in the inherent variability introduced by the API-based inference process. To explore the upper-bound performance under identical settings (same random seed), we perform multiple runs for each experiment and report best results. However, similar to other prompt optimization methods, this stochasticity cannot be fully eliminated and may still influence performance evaluation. Additionally, due to the relatively small number of validation and test examples (consistent with prior work on prompt optimization) there is a risk of overfitting to the validation set in certain runs. Future work may address these challenges by incorporating more robust evaluation protocols and exploring prompt tuning strategies that are less sensitive to sampling noise.

In addition, our method can enhance the performance of LLMs on domain-specific tasks, thereby enabling more effective decision support (e.g., identifying turning points in pandemic trends). However, the reasoning processes of LLMs may produce incorrect or unreliable results for certain cases. Future research could focus on improving the trustworthiness of LLMs in such domain-specific applications.

## 8.7 Example of Organized Prompts

---

**Prompt 8.2: Example of Organized Prompts for Pandemic Dataset**

[System Prompt] Predict the trend of hospitalizations for the next week based on the pandemic details provided between `<Pandemic Description>` and `<\Pandemic Description>`.

Provide a single prediction enclosed in `< >` using one of the following labels: `<substantial decreasing>`, `<moderate decreasing>`, `<stable>`, `<moderate increasing>`, `<substantial increasing>`.

Definitions:

– "Substantial" refers to changes greater than 3.

– "Moderate" corresponds to changes between 1 and 3.

– "Stable" is defined as changes between -1 and 1.

The final line of your response must follow this format: `<VALUE>`, where `VALUE` is your prediction.

`<Pandemic Description>`

[Demographic Information] Vermont, with one of the smallest populations and one of the smallest Black demographic groups, voted Democratic in the recent Presidential election.

[Healthcare System Condition] During the pandemic, Vermont's healthcare systems performed among the best, with above-national-average Access and Affordability, excellent Prevention and Treatment, better-than-average population health conditions, and reduced Income Disparity.

[ICU and Hospital Staffing Condition] Vermont had ICU stress levels near the national average, but hospital staffing shortages worse than the national average.

[Vaccination Coverage] As of now, 81% of the population has received at least one vaccine dose (Rapid Increase trend), 71% are fully vaccinated (Moderate Increase trend), and 23% received boosters (Rapid Increase trend).

[Population Immunity] Around 28% of the population reported infections in the past three months, and population immunity is showing a Rapid Increase.

[Restriction Policy Response] School closures were recommended, but there were no restrictions for workplaces or gatherings among elderly patients. Isolation was recommended, and visitor restrictions were in place.

[Hospitalization per 100k] The average number of COVID-19 hospitalizations per 100K over the past five weeks was 9.8. Hospitalizations remained relatively stable, mostly between 9.0 and 11.2. A slight increase was observed in the most recent week, with a rate of change of 1.2. Volatility in hospitalization numbers was minimal, indicating consistent trends.

[Reported Cases per 100k] In the most recent five weeks, reported COVID-19 cases per 100K showed a fluctuating trend. The average was 292.6. Cases declined from 263.8 to 216.0 over the first three weeks, then sharply increased to 340.1 in the fourth week and 398.7 in the fifth. These changes indicate a significant uptick in recent weeks, with inconsistent weekly trends.

`<\Pandemic Description>`

---

[Trend of Hospitalization (Ground Truth)] `<moderate increasing>`

**Prompt 8.3: Example of Organized Prompts for TrafficSafe Dataset**

**[System Prompt]** Predict the crash severity reasoning on the causal descriptions provided between the crash event details provided between `<Crash Description>` and `<\Crash Description>`.
Provide a single prediction enclosed in `< >` using one of the following labels:
`<no apparent injury>`, `<minor injury>`, `<serious injury>`, `<fatal>`.
The last line of your response should only be of the following format: `<VALUE>` where `VALUE` is your prediction.

`<Crash Description>`
**[Time]** The crash occurred on April 29, 2022 at hour 16.

**[Position]** The crash occurred in Champaign, within an Unincorporated area. It did not occur in a work zone.

**[Dynamic Conditions]** The light condition is Daylight and the weather condition is Clear.

**[Infrastructure]** The crash is not at an intersection. The traffic control device is Other Regulatory Sig.

**[Road Surface]** The road surface condition is Dry. The road defect condition is nan.

**[Road Level]** The trafficway is Not Divided Two-way. The functional class of the roadway is Minor Arterial. The roadway class is Rural 2 Lane Roads.

**[Driver Behavior]** The primary behavior is Driving On Wrong Side/Wrong Way, and the secondary behavior is Improper Lane Usage. The crash is not a hit-and-run incident.

**[Vehicle 1 Vehicle Information]** The vehicle had a defect of None and was manufactured in 2004.
**[Vehicle 2 Vehicle Information]** The vehicle had a defect of None and was manufactured in 2002.

**[Vehicle 1 Vehicle Status]** The unit locates at On Pavement (Roadway). The vehicle's maneuver prior to the crash was Passing/Overtaking and it was traveling in the North direction.
**[Vehicle 2 Vehicle Status]** The unit locates at On Pavement (Roadway). The vehicle's maneuver prior to the crash was Straight Ahead and it was traveling in the South direction.

**[Person 1 Person Information]** This person was in Vehicle Unit 1. The person involved is a Driver, aged 39. Gender is Male.
**[Person 2 Person Information]** This person was in Vehicle Unit 2. The person involved is a Driver, aged 70. Gender is Male.

**[Person 1 Person Status]** The driver's blood alcohol content is .000. Distraction status: No. Safety equipment used: Shoulder and Lap Belt Used.
**[Person 2 Person Status]** The driver's blood alcohol content is Not Tested. Distraction status: No. Safety equipment used: Shoulder and Lap Belt Used.
`<\Crash Description>`

---

**[Crash Severity (Ground Truth)]** `<fatal>`

**Prompt 8.4: Example of Organized Prompts for Swissmetro Dataset**

[System Prompt] Predict the travel mode choice reasoning on the causal descriptions pro-
vided between <Causal Description> and <\Causal Description>, and the traveler
details provided between <Traveler Description> and <\Traveler Description>.
Provide a single prediction enclosed in < > using one of the following labels:
<swissmetro>, <car>, <train>.
The final line of your response must follow this format: <VALUE>, where VALUE is your
prediction.

<Traveler Description>
[**trip_purpose**] The purpose of the trip is business.
[**trip_paid_by**] Traveler trip is paid by oneself.
[**luggage**] Traveler has no luggage.

[**first_class**] The traveler earns 100,000 CHF and does not travel in first class.
[**rail_pass**] Traveler does not have a rail-system annual season ticket.

[**origin_destination**] This trip starts at VD and ends at ZH.
[**options_count**] Traveler has two possible travel options.

[**swissmetro_time_cost**] Swissmetro's travel time is 63 minutes and it costs 57 CHF.
[**swissmetro_headway**] The headway of Swissmetro is 10 minutes.
[**train_time_cost**] Train's travel time is 192 minutes and it costs 52 CHF.
[**train_headway**] The headway of train is 30 minutes.

[**income**] Traveler's annual income is between 50,000 and 100,000 CHF.
[**age_range**] The traveler is between 39 and 54 years old.
[**gender**] The traveler is female.
<\Traveler Description>

---

[Travel Mode Choice (Ground Truth)] <car>

## 8.8 Initial SCG

---

**Prompt 8.5: Initial SCG for Pandemic Dataset**

CAUSAL STATEMENT 1: **[Demographic Information]** affects **[Vaccination Coverage]** and **[Restriction Policy Response]**.
Older or vulnerable populations often have higher vaccination uptake and are more likely to be targeted by stricter restrictions.

CAUSAL STATEMENT 2: **[Healthcare System Condition]** affects **[Vaccination Coverage]** and **[Population Immunity]**.
Regions with better healthcare access can distribute vaccines more effectively and maintain higher baseline immunity.

CAUSAL STATEMENT 3: **[ICU and Hospital Staffing Condition]** affects **[Restriction Policy Response]**.
When ICU beds are full or staffing is limited, governments tend to implement stricter control policies.

CAUSAL STATEMENT 4: **[Vaccination Coverage]** affects **[Population Immunity]**.
Higher vaccination coverage directly increases the proportion of immune individuals in the population.

CAUSAL STATEMENT 5: **[Population Immunity]** affects **[Reported Cases per 100k]** and **[Hospitalization per 100k]**.
Stronger immunity reduces both the number of new infections and the chance of severe cases needing hospitalization.

CAUSAL STATEMENT 6: **[Reported Cases per 100k]** affects **[Hospitalization per 100k]** and **[Restriction Policy Response]**.
A rise in reported cases usually precedes more hospital admissions and can trigger policy tightening.

CAUSAL STATEMENT 7: **[Hospitalization per 100k]** affects **[Restriction Policy Response]**.
High hospitalization levels often lead to immediate government intervention to limit further spread.

CAUSAL STATEMENT 8: **[Hospitalization per 100k]** and **[Restriction Policy Response]** affect **[Change of Hospitalization Next Week]**.
The trends of hospitalization in past weeks have strong relation with change of hospitalization next week.

---

**Prompt 8.6: Initial SCG for TrafficSafe Dataset**

CAUSAL STATEMENT 1: **[Person Status]** affects **[Severity]**.
The driver's Blood Alcohol Content (BAC) significantly increases the probability of fatal crashes.

CAUSAL STATEMENT 2: **[Position]** affects **[Severity]**.
Work zones can increase the probability of serious and fatal crashes. Driving in work zones after drinking is especially likely to cause severe or fatal crashes.

CAUSAL STATEMENT 3: **[Driver Behavior]** affects **[Severity]**.
Aggressive driving and impairment-related behavior pose higher risk than other driver behaviors.

---

**Prompt 8.7: Initial SCG for Swissmetro Dataset**

CAUSAL STATEMENT 1: **[Gender]** and **[Age]** affect **[Trip Purpose]** and **[Luggage]**.
Younger travelers are more likely to travel for education or leisure and carry luggage; older travelers more often travel for business with less luggage.

CAUSAL STATEMENT 2: **[Income]** affects **[First Class]**, **[Rail Pass]**, and **[Trip_Paid_By]**.
High-income travelers are more likely to choose first class, own a rail pass, and pay for the trip themselves.

CAUSAL STATEMENT 3: **[Trip Purpose]** affects **[Trip_Paid_By]** and **[Luggage]**.
Business trips are often employer-paid and involve less luggage; leisure trips are usually self-paid and involve more.

CAUSAL STATEMENT 4: **[Origin and Destination]** determine **[Travel Options]**, **[Travel Time]**, and **[Headway]**.
Major city pairs offer more modes, shorter travel time, and higher frequency.

CAUSAL STATEMENT 5: **[Trip Purpose]** affects **[Travel Mode Choice]**.
Business travelers tend to prefer faster, more reliable modes; leisure travelers may prioritize cost or flexibility.

CAUSAL STATEMENT 6: **[First Class]** affects **[Travel Mode Choice]**.
Travelers choosing first class are more likely to select Train or Swissmetro over Car for comfort.

CAUSAL STATEMENT 7: **[Rail Pass]** affects **[Travel Mode Choice]**.
Travelers with a rail pass are more likely to use Train or Swissmetro due to lower perceived cost.

CAUSAL STATEMENT 8: **[Luggage]** affects **[Travel Mode Choice]**.
Travelers with heavy or bulky luggage may prefer Train or Car.

CAUSAL STATEMENT 9: **[Trip_Paid_By]** affects **[Travel Mode Choice]**.
If the trip is employer-paid, travelers tend to choose faster or more comfortable modes like Swissmetro; if self-paid, they prefer cheaper options like standard Train or Car.

CAUSAL STATEMENT 10: **[Travel Time]** and **[Headway]** affect **[Travel Mode Choice]**.
Business travelers are more sensitive to time and prefer faster and frequent modes; leisure travelers may tolerate longer travel time or wait if the mode is cheaper or more flexible.

## 8.9  Refined SCG

---

**Prompt 8.8: Refined SCG for Pandemic Dataset (Compared to Initial)**

CAUSAL STATEMENT 1: **[Demographic Information]** affects **[Vaccination Coverage]** ~~and~~ ~~**[Restriction Policy Response]**~~.
Older or vulnerable populations often have higher vaccination uptake. ~~and are more likely to be targeted by stricter restrictions.~~

CAUSAL STATEMENT 2: **[Healthcare System Condition]** ~~affects **[Vaccination Coverage]** and~~ affects **[Population Immunity]**.
~~Regions with better healthcare access can distribute vaccines more effectively and~~ maintain higher baseline immunity.

CAUSAL STATEMENT 3: **[ICU and Hospital Staffing Condition]** affects **[Restriction Policy Response]**.
When ICU beds are full or staffing is limited, governments tend to implement stricter control policies.

CAUSAL STATEMENT 4: **[Vaccination Coverage]** affects **[Population Immunity]**.
Higher vaccination coverage directly increases the proportion of immune individuals in the population.

CAUSAL STATEMENT 5: **[Population Immunity]** affects **[Reported Cases per 100k]** ~~and~~ ~~**[Hospitalization per 100k]**~~.
~~Stronger immunity reduces both the number of new infections and the chance of severe cases needing hospitalization.~~ Stronger immunity reduces the number of new infections.

CAUSAL STATEMENT 6: **[Reported Cases per 100k]** affects **[Hospitalization per 100k]**. ~~and~~ ~~**[Restriction Policy Response]**~~
A rise in reported cases usually precedes more hospital admissions. ~~and can trigger policy tightening.~~

CAUSAL STATEMENT 7: **[Hospitalization per 100k]** affects **[Restriction Policy Response]**.
High hospitalization levels often lead to immediate government intervention to limit further spread.

CAUSAL STATEMENT 8: **[Hospitalization per 100k]** and **[Restriction Policy Response]** affect **[Change of Hospitalization Next Week]**.
The trends of hospitalization in past weeks have strong relation with change of hospitalization next week.

CAUSAL STATEMENT 9: **[Healthcare System Condition]** affects **[Hospitalization per 100k]**.
Poor healthcare system performance can lead to higher hospitalization rates due to inadequate prevention and treatment measures.

---

**Prompt 8.9: Refined SCG for TrafficSafe Dataset (Compared to Initial)**

CAUSAL STATEMENT 1: **[Person Status]** affects **[Severity]**.
The driver's Blood Alcohol Content (BAC) will significantly increase the probability of
`<FATAL>` crashes.

CAUSAL STATEMENT 2: **[Position]** affects **[Severity]**.
Work zone can increase the probability of `<SERIOUS INJURY>` and `<FATAL>` crashes. Driving in a work zone after drinking is very likely to cause `<SERIOUS INJURY>` or `<FATAL>`
crashes.

CAUSAL STATEMENT 3: **[Driver Behavior]** affects **[Severity]**.
Aggressive driving and impairment-related behavior are more risky than other driver behaviors.

CAUSAL STATEMENT 4: **[Road Conditions]** affects **[Severity]**.
Icy road conditions can significantly increase the likelihood of `<SERIOUS INJURY>` or
`<FATAL>` outcomes due to loss of vehicle control.

CAUSAL STATEMENT 5: **[Safety Equipment]** affects **[Severity]**.
The use of safety equipment, such as shoulder and lap belts, is likely to reduce the severity of
injuries in the event of a crash.

CAUSAL STATEMENT 6: **[Road Level]** affects **[Severity]**.
Two-way undivided roads can increase the likelihood of `<SERIOUS INJURY>` or `<FATAL>`
outcomes due to the potential for head-on collisions.

CAUSAL STATEMENT 7: **[Dynamic Conditions]** affects **[Severity]**.
Daylight and clear weather conditions are generally associated with lower crash severity,
reducing the likelihood of `<SERIOUS INJURY>` or `<FATAL>` outcomes.

CAUSAL STATEMENT 8: **[Infrastructure]** affects **[Severity]**.
The presence of traffic control devices, such as stop signs, can influence crash severity by
regulating vehicle flow and reducing collision risks.

CAUSAL STATEMENT 9: **[Vehicle Information]** affects **[Severity]**.
Older vehicles or those with unknown defects may contribute to higher crash severity due to
potential safety feature limitations.

**Prompt 8.10: Refined SCG for Swissmetro Dataset (Compared to Initial)**

CAUSAL STATEMENT 1: **[Gender]** and **[Age_Range]** affect **[Trip Purpose]** and **[Luggage]**.
Younger travelers are more likely to travel for education or leisure and carry luggage; older travelers more often travel for business with less luggage.

CAUSAL STATEMENT 2: **[Income]** affects **[First_Class]** ~~, [rail pass]~~ and **[Self-Paid]**.
high-income travelers are more likely to choose first class, own a rail pass, and pay for the trip themselves.

CAUSAL STATEMENT 3: **[Trip Purpose]** affects **[Self-Paid]** and **[Luggage]**.
Business trips are often employer-paid and involve less luggage; leisure trips are usually self-paid and involve more.

~~CAUSAL STATEMENT 4: [Origin and destination] determine [travel options], [travel time], and [headway]~~
~~(major city pairs offer more modes, shorter travel time, and higher frequency)~~

~~CAUSAL STATEMENT 5: [Trip purpose] affects [travel mode choice]~~
~~(business travelers tend to prefer faster, more reliable modes; leisure travelers may prioritize cost or flexibility)~~

~~CAUSAL STATEMENT 6: [First class] affects [travel mode choice]~~
~~(travelers choosing first class are more likely to select Train or Swissmetro over Car for comfort)~~

CAUSAL STATEMENT 4: **[Rail Pass]** affects **[Travel Mode Choice]**.
Travelers with a rail pass are more likely to use Train or Swissmetro due to lower perceived cost.

~~CAUSAL STATEMENT 8: [Luggage] affects [travel mode choice]~~
~~(travelers with heavy or bulky luggage may prefer Train or Car)~~

CAUSAL STATEMENT 5: **[Trip_Paid_By]** affects **[Travel Mode Choice]**.
If the trip is employer-paid, travelers tend to choose faster or more comfortable modes like Swissmetro; if self-paid, they prefer cheaper options like standard Train or Car.

CAUSAL STATEMENT 6: **[Travel Time]** and **[Headway]** affect **[Travel Mode Choice]**.
Business travelers are more sensitive to time and prefer faster and frequent modes; leisure travelers may tolerate longer travel time or wait if the mode is cheaper or more flexible.

## 8.10 Constraints for SCG Optimizer

---

**Prompt 8.11: Constraints for SCG Optimizer for Pandemic Dataset**

**Prompt Format**
Your revised prompt must follow the structure below:

- `<SYSTEM PROMPT>`
  (system prompt)
  `<\SYSTEM PROMPT>`
- `<Causal Relations>`
  (causal relations)
  `<\Causal Relations>`
- `<Output>`
  (output format, fixed, don't revise this)
  `<\Output>`

**Causal Relations Guidelines**
Only include causal relations between nodes for which corresponding information is available in the input description:

- `[Demographic Information]`
- `[Healthcare System Condition]`
- `[ICU and Hospital Staffing Condition]`
- `[Vaccination Coverage]`

- `[Population Immunity]`
- `[Restriction Policy Response]`
- `[Hospitalization per 100k]`
- `[Reported Cases per 100k]`

**Operations**
You can only use the following operations:
**[1] Add** new causal relations if they are clearly supported by the input. Do not make assumptions without evidence. Use the format:

> `[Node A] affects [Node B]`
> (Explanation of how [Node A] affects [Node B])

Both `[Node A]` and `[Node B]` must come from the list above. You may include node-to-node relations not involving the final prediction target to support broader reasoning and imputation.

**[2] Modify** existing causal relations. You may:

- Replace `[Node A] affects [Node B]` with a more accurate link such as `[Node A] affects [Node C]`
- Update the explanation for clarity or correctness

**[3] Delete** any causal relation that is unsupported or may negatively impact model inference. Remove both the relation and its explanation.

---

