# OpenReview forum: "How to Auto-optimize Prompts for Domain Tasks? Adaptive Prompting and Reasoning through Evolutionary Domain Knowledge Adaptation"
_NeurIPS.cc/2025/Conference — NeurIPS 2025 poster_

### Official Review · Reviewer_qzNs · 2025-06-30

**Clarity:** 3
**Significance:** 3
**Originality:** 3
**Rating:** 4
**Confidence:** 3

**Summary:**

- This paper introduces EGO-Prompt, a novel framework for optimizing domain-specific reasoning in LLMs by integrating expert-defined semantic causal graphs (SCGs) and evolving them via textual gradients. The method decomposes reasoning into guidance generation and prediction stages and efficiently reduces inference costs.

**Questions:**

- I found that the LLMs used in the experiments are all closed APIs like GPT-40-mini and Gemini-2.5-Flash. These LLMs are mature commercial products with better instruction-following ability. Can EGO-Prompt be applied on open-source LLM backbones like llama or Qwen? For domain-specific application, it seems that privatized deployment of small-scale LLM is a necessity and open-source LLMs might be a better choice.
- Does EGO-Prompt have cross-domain generalization ability? For example, train EGO-Prompt on one domain and generalize to another similar but not equal domain.
- What is the cost for manual design of SGGs? Can SGG also be constructed automatically by the LLMs?

**Ethical Concerns:**

["NO or VERY MINOR ethics concerns only"]

**Final Justification:**

I'd like to keep positive score as authors make enough response to my questions.

**Limitations:**

Yes

**Quality:**

3

**Strengths And Weaknesses:**

## Strengths
- EGO-Prompt's two-stage reasoning process—generating instance-specific guidance from SCGs and optimizing prompts via textual gradients effectively bridges domain knowledge with LLM capabilities. This is a novel prompt engineering design.
- The framework enables lightweight models (e.g., GPT-4o mini) to achieve or surpass state-of-the-art performance.

## Weaknesses
- The framework assumes human experts provide initial SCG, introducing potential biases that optimization may not fully correct. This design gap is not rigorously tested against imperfect expert inputs, weakening the method's autonomy and real-world applicability where expert knowledge is sparse or noisy.
- The datasets used in experiments are relatively small-scale, and the number of specific domain is limited.
- See questions for more question about this paper.

---

> ### Author Rebuttal · Authors · 2025-07-30
>
> >  **W1: Lack of test with imperfect expert inputs.**
>
> Thank you for the constructive comment. We fully agree that imperfect expert inputs are common in real-world applications, and it is important to evaluate the robustness of our method under such conditions. To evaluate this, we have conducted additional experiments where we evluate our model's performance under imperfect expert inputs. Please see response to **W3 of Reviewer KdBz**
>
> >  **W2: The datasets are small-scale, and the number of specific domain is limited.**
>
> Thank you for the comment.
>
> - **We follow the standard settings used in discrete prompt optimization methods.** The commonly used benchmarks are GSM8K [1], a dataset of 7,473 training and 1,319 test samples of grade-school math word problems, and Big-Bench Hard (BBH) [2], which consists of 23 reasoning tasks with up to 250 examples per task, covering symbolic, logical, and commonsense reasoning. For BBH, during prompt optimization, each task is typically split into training, validation, and testing sets, with around 80 samples in each split. This is the common settings for those prompt optimization methods like Textual Gradient [3]. In our evaluation, we use datasets from three different domain, each containing approximately 300 samples for optimization, which is comparable in scale to existing benchmarks and methods.
>
> - **Our prompt length is substantially longer than commonly used benchmarks.** The average prompt length in GSM8K and BBH is around 100 tokens per question. In contrast, our datasets feature an average prompt length of approximately 500 tokens, incorporating much more complex contextual information and interactions among multiple factors.
>
> - **We cover multiple domains with real-world complexity.** While existing benchmarks often focus on a single type of reasoning task (e.g., given A and B, what is C), our datasets span three distinct domains: public health (Pandemic), traffic safety (TrafficSafe), and human behavior (Swissmetro). These domains involve diverse data structures, contextual dependencies, and decision factors. We also plan to extend our method to additional real-world application areas in future work.
>
>
>
> > **Q1: Can EGO-Prompt be applied on open-source LLM?**
>
> Thank you for the helpful comment. In addition to the commercial models GPT-4.1-mini, GPT-4o-mini, Gemini 2.5 Flash, and Gemini 2.0 Flash, we extended our experiments to evaluate EGO-Prompt on open-source models, including Qwen3 (1.7B, 8B, 14B, 32B), DeepSeek-V3, and Llama-4-Scout-17B. We summarize several findings below:
>
> |Type|BaseModel|PandemicAcc|PandemicF1|TrafficSafeAcc|TrafficSafeF1|SwissmetroAcc|SwissmetroF1|MeanF1(improvements)|
> |---|---|---|---|---|---|---|---|---|
> |**Open models**|Qwen-1.7B(Base)|0.120|0.158|0.110|0.120|0.047|0.084|0.121|
> ||+EGO-Prompt|0.190|0.178|0.200|0.230|0.412|0.386|**0.265(↑119.5%)**|
> ||Qwen-8Bc(Base)|0.060|0.080|0.220|0.100|0.059|0.096|0.092|
> ||+EGO-Prompt|0.140|0.199|0.290|0.202|0.459|0.427|**0.276(↑200.1%)**|
> ||Qwen-14B (Base)|0.360|0.299|0.330|0.227|0.424|0.309|0.279|
> ||+EGO-Prompt|0.390|0.339|0.340|0.271|0.471|0.457|**0.356(↑27.7%)**|
> ||Qwen-32B (Base)|0.340|0.284|0.310|0.229|0.412|0.321|0.278|
> ||+EGO-Prompt|0.430|0.405|0.350|0.285|0.494|0.481|**0.390(↑40.3%)**|
> ||DeepSeekV3 (Base)|0.210|0.231|0.340|0.211|0.459|0.457|0.300|
> ||+EGO-Prompt|0.460|0.464|0.400|0.371|0.506|0.501|**0.445(↑48.7%)**|
> ||Llama-4-Scout-17B-16E-Instruct-FP8(Base)|0.360|0.307|0.260|0.237|0.294|0.287|0.277|
> ||+EGO-Prompt|0.420|0.376|0.320|0.268|0.482|0.459|**0.368(↑33.0%)**|
> |**Close models**|GPT-4o-mini (Base)|0.360|0.347|0.300|0.232|0.459|0.406|0.328|
> ||+EGO-Prompt|0.410|0.399|0.380|0.333|0.506|0.498|**0.410(↑24.9%)**|
> ||Gemini-2.5-Flash (Base)|0.470|0.470|0.340|0.319|0.459|0.392|0.394|
> ||+EGO-Prompt|0.540|0.546|0.430|0.428|0.518|0.499|**0.491(↑24.6%)**|
> ||GPT-4.1-mini (Base)|0.400|0.374|0.350|0.246|0.506|0.468|0.363|
> ||+EGO-Prompt|0.430|0.420|0.370|0.292|0.553|0.522|**0.411(↑13.2%)**|
> ||Gemini-2.0-Flash (Base)|0.300|0.268|0.330|0.264|0.259|0.351|0.294|
> ||+EGO-Prompt|0.430|0.412|0.400|0.327|0.506|0.470|**0.403(↑37.0%)**|
>
> - **EGO-Prompt is effective for open source models.** Excluding Qwen3-1.7B and Qwen3-8B, EGO-Prompt improves the performance of open-source models by 27.7%–48.7%, comparable to its improvements on commercial models.
>
> - **Commercial models achieve better and more stable performance.**  Across both the initial organized prompt and the optimized settings, commercial models consistently outperform open-source models, achieving higher F1 scores and greater robustness across tasks.
>
> - **Model size substantially influences the performance ceiling.** We can find the model with more parameters achieve better optmizaed performance, consistent with the scaling law. For example, the Qwen3 series improves from an F1 score of 0.265 for the 1.7B model to 0.390 for the 32B model. DeepSeek-V3, with 671B parameters, achieves the highest performance among the open-source models. These results suggest that our method can consistently enhance performance across models of varying sizes.
>
> - **EGO-Prompt enhances reasoning in smaller models.** Models such as Qwen3-1.7B and Qwen3-8B fail to generate structured outputs under the initial organized prompt. However, with EGO-Prompt, they are able to produce more structured predictions and follow certain reasoning paths to make predictions, although their overall performance remains limited.
>
> **In conclusion, commercial models are a better choice for our method.** Motivated by the need to better integrate expert knowledge into the LLM reasoning process, we design our framework to minimize deployment complexity and maximize usability for non-technical users. Since most end-users may lack the infrastructure to run models locally, our key focus is on improving the reasoning performance of black-box LLMs for domain-specific tasks. As commercial models typically exhibit better instruction-following ability, we recommend using lightweight commercial LLMs as the forward engine, although our method is compatible with both open-source and closed-source models.
>
> We have revised our manuscript to incorporate the above results and discussion into the appendix.
>
> >  **Q2: Does EGO-Prompt have cross-domain generalization ability?**
>
> Thanks! And the answer is yes in real-world applications. We tested the EGO Prompt method across three different real-world tasks (transportation, human choice and public health), achieving clear performance and cost advantages. More specifically, our method demonstrates generalization ability in the following two aspects:
>
> - **Domain-level Generalization.** We evaluate generalization by directly applying the SCG and system prompt optimized on one dataset to another, testing the model’s ability to transfer across domains. In the TrafficSafe dataset, we predict crash severity using crash report data from the HSIS [4]. Our original model was optimized on data from Illinois, which we now adapt to Washington State. These two datasets differ in several respects. For instance, Washington includes features such as daily road traffic volume, which are not available in the Illinois dataset. Additionally, the distribution of crash severity varies across states ,as Washington has fewer fatal and serious injury cases compared to Illinois.
>
>   The table below summarizes the performance on the Washington dataset using the SCG and system prompt optimized on the Illinois dataset. We uniformly sampled 100 instances from the Washington dataset across all crash severity classes. GPT-4o-mini and Gemini 2.5 Flash were used as the forward engine. We tested three configurations: replacing the initial SCG, the system prompt, or both with those optimized on the Illinois dataset, and recorded the corresponding performance on the test set. The results indicate that transferring either the SCG or the system prompt individually leads to performance gains, while combining both yields the most substantial improvement.
>
> | Setting                                  | GPT-4o-mini Acc | GPT-4o-mini F1 | Gemini 2.5 Flash Acc | Gemini 2.5 Flash F1 |
> | -- | ---- | --- | -- | -- |
> | Organized Prompt              | 0.300           | 0.222          | 0.350                | 0.312               |
> | IL SCG Only                                 | 0.340           | 0.251          | 0.390                | 0.353               |
> | IL System Prompt Only                       | 0.330           | 0.272          | 0.410                | 0.365               |
> | IL SCG + System Prompt | **0.340**       | **0.285**      | **0.440**            | **0.401**           |
>
> - **Task-level Generalization.** Our method is easily adaptable to new tasks, requiring only the redesign of the SCG. As shown in Table 3 of our manuscript, EGO-Prompt consistently improves reasoning performance across diverse tasks.
>
> >  **Q3: What is the cost for manual design of SGGs? Can SGG also be constructed automatically by the LLMs?**
>
> Thank you for the comment. The manual design of SCGs requires minimal human effort, typically less than 10 minutes for a domain expert familiar with the task and data. Since EGO-Prompt is capable of automatically refining and correcting SCGs during the optimization process, the initial SCG does not need to be perfectly accurate (see our response to W1). This substantially reduces the cost and burden of manual design.
>
>  SCGs can also be constructed automatically by LLMs. Please refer to our response to W1 for details on SCG generation starting from a null initialization.
>
>
> ## References
> [1] Cobbe, K., et al. Training verifiers to solve math word problems. arXiv, 2021.
>
> [2] Suzgun, M. et al. Challenging big-bench tasks and whether chain-of-thought can solve them. Findings of ACL. 2022.
>
> [3] Yuksekgonul, M., et al. Optimizing generative ai by backpropagating language model feedback. Nature. 2025.
>
> [4] U.S. Department of Transportation, Federal Highway Administration. Highway Safety Information System.

---

> > ### Author Response · Authors · 2025-08-06
> >
> > Dear Reviewer qzNs,
> >
> > Thank you for your valuable comments. Your suggestions, especially Q1 and Q2, have greatly helped us improve the quality of our paper. We have conducted additional experiments to address your questions:
> >
> > **Llama 3.3 models results.**
> > |Type|BaseModel|PandemicAcc|PandemicF1|TrafficSafeAcc|TrafficSafeF1|SwissmetroAcc|SwissmetroF1|MeanF1(improvements)|
> > |---|---|---|---|---|---|---|---|---|
> > |**Open Source**| LLaMA 3.3 8B (Base)  | 0.350       | 0.274       | 0.310           | 0.210          | 0.310          | 0.210          | 0.231                   |
> > |              | +EGO-Prompt        | 0.370       | 0.326       | 0.330           | 0.289          | 0.377          | 0.335          | **0.316 (↑36.8%)**      |
> > |              | LLaMA 3.3 70B (Base) | 0.370       | 0.317       | 0.250           | 0.176          | 0.400          | 0.275          | 0.256                   |
> > |              | +EGO-Prompt        | 0.420       | 0.378       | 0.340           | 0.273          | 0.459          | 0.419          | **0.357 (↑39.5%)**      |
> >
> > We look forward hearing your feedback and, we will be more than happy learn from your comments and thoughts!
> >
> > Sincerely,
> >
> > Authors 8465

---

> ### Author Response · Authors · 2025-08-08
> **Updates on GPT-5-mini Results**
>
> Dear Reviewer qzNs,
>
> Following the release of OpenAI’s new GPT-5 series models, we conducted additional experiments to evaluate our method on the GPT-5-mini (minimal reasoning) model. The results are shown below:
>
>
> | Setting          | Pandemic Acc | Pandemic F1 | TrafficSafe Acc | TrafficSafe F1 | Swissmetro Acc | Swissmetro F1 | Mean F1 (Improvement) |
> | ---------------- | ------------ | ----------- | --------------- | -------------- | -------------- | ------------- | --------------------- |
> | Organized Prompt | 0.420        | 0.387       | 0.340           | 0.260          | 0.388          | 0.390         | 0.346                 |
> | EGO-Prompt       | **0.460**    | **0.448**   | **0.340**       | **0.305**      | **0.529**      | **0.511**     | **0.421 (+21.9%)**    |
>
> Please let us know your thoughts on our response and revisions. We would be more than happy to discuss further if you have any questions!
>
> Authors, 8465

---

### Official Review · Reviewer_UAca · 2025-06-30

**Clarity:** 3
**Significance:** 3
**Originality:** 3
**Rating:** 4
**Confidence:** 3

**Summary:**

The paper proposes the Evolutionary Graph Optimization for Prompting (EGO-Prompt), an automated framework designed to craft optimal prompts, refine reasoning processes, and enhance causal inference. The framework starts with a general prompt constructed by human experts and an initial fault-tolerant Semantic Causal Graph (SCG) description, then guides LLM reasoning through iterative automated optimization.
The framework was tested on real-world tasks in public health, transportation, and human behavior. Results show that EGO-Prompt achieves an 8%-12% improvement in F1-score compared to state-of-the-art methods while generating refined domain-specific SCGs.

**Questions:**

1. How much extra inference cost does it require compared to ​Zero-shot-CoT​ or ​Organized Prompt​?

2. The baseline methods compared in the paper do not include other graph-based approaches. Are there such baselines available?

3. How does it perform against automated ICL methods like ​AutoCoT​?

**Ethical Concerns:**

["NO or VERY MINOR ethics concerns only"]

**Final Justification:**

The authors' experiments demonstrate its low computational cost and strong performance across multiple models, so I will raise my score.

**Limitations:**

yes

**Paper Formatting Concerns:**

No Formatting problem

**Quality:**

3

**Strengths And Weaknesses:**

Strengths:

The paper proposes a novel framework that employs a forward-backward simulation process to iteratively optimize the SCG and instance-specific prompts.
This approach generates more refined domain-specific prompts, enabling LLMs to enhance their reasoning steps and improve performance in specialized domains.

Weaknesses:
1. The method proposed in the paper appears to be an optimization based on TextGrad combined with some SCG techniques—the approach is relatively simple;

2. The additional computational overhead incurred during the optimization process remains unclear.

3. The results show that ProTeGi (GPT-4o mini) achieves higher accuracy than the proposed method in certain settings. Additionally, the performance gap between Zero-shot-CoT and EGO is relatively small.

---

> ### Author Rebuttal · Authors · 2025-07-30
>
> > **W1: The approach is relatively simple.**
>
> Thank you for the comment. Our method innovatively integrate domain expert knowledge into LLMs for better reasoning in domain-specific tasks. Our contribution includes
>
> - **Expert knowledge guided LLM raesoning in domain-specific tasks.** Existing methods for domain-specific reasoning often rely on external graph databases to retrieve structured knowledge. However, this database is often unavailable in real-world scenarios. We use SCG to encode expert knowledge which not only provides explicit causal relations to guide LLM reasoning but also evolves during the optimization process, making the method adaptive and dynamic.
>
> - **Lossless instance-specific reasoning guidance which aligns the structural graph information with textual information.** Our experiments reveal that directly feeding SCGs into a single LLM yields degraded performance (see Table 4). This suggests that graph-structured information, such as SCGs, is not directly compatible with LLM inputs in its raw form. To better integrate graph information into natural language prompts, we introduce a causal description model (see Figure 2) that aligns SCGs with natural language. By satisfying the Conditional Independence and Deterministic Guidance conditions (see Section 8.1), the model produces lossless textual descriptions that effectively preserve the high-order dependencies encoded in the graph while making them accessible to language models.
>
> - **Overall Improvement in performance, cost-efficiency, cross-domain generalization, interpretability, and the co-pilot design.** Please see response to W3.
>
> - **Textual Gradient is only used as the iteration optimizer, like Adam. We also include it as a baseline method.** Please refer to the response to W2 of Reviewer DhWo for details.
>
> **Being simple and effective is a merit, especially in real-world tasks.** Our method is a multimodal prompt optimization framework that incorporates graph-based priors for real-world domain tasks (e.g., transportation, public health, and human behavior modeling) that require low cost, high interpretability, and strong reasoning quality. Through careful system design and thorough empirical validation, we achieve clear improvements in these aspects (**we also added 6 more experiments on open-source LLMs, see the long table in reviewer qzNs**). Our lightweight, plug-and-play method offers a practical solution with merit that can be readily adopted in real-world decision-making systems.
>
> >  **W2: The additional computational overhead incurred during the optimization process is unclear.**
>
> Thank you for raising this important point. We have carefully evaluated the cost of our method throughout the optimization process, using GPT-4o-mini as the forward engine and GPT-4o as the backward engine. Each optimization step includes two forward passes through both the graph description model and the prediction model, two backward passes to iteratively update the SCG, system prompt, and causal system prompt, as well as two evaluations on the validation set and zero or one evaluation on the test set (see Algorithm 1). We recorded the actual costs using the OpenAI API platform during the optimization and reported the average cost over three steps for each task.
>
> |Dataset|Opt. Cost per Step (USD)|
> |----|----|
> |Pandemic|0.310|
> |TrafficSafe|0.403|
> |Swissmetro|0.313|
>
> As each task typically requires 6 to 12 optimization steps, the total cost for the full optimization process ranges from approximately \\$2 to \\$5 per task. We have added this information to the appendix for reference.
>
> >  **W3: ProTeGi (GPT-4o mini) achieves higher accuracy than the proposed method in certain settings. The Performance gap between Zero-shot-CoT and EGO is relatively small.**
>
> Thank! We understand your concern regarding the performance improvements. However, given that the performance of LLMs is generally constrained by **scaling laws**, we would like to clarify that our contributions extend beyond prediction accuracy and also include improvements in **cost-efficiency, cross-domain generalization, interpretability, and the co-pilot design**.
>
> - **Cost-efficiency.** As shown in Figure 4 and Table 7, we make **GPT-4o-mini comparable with reasoning models like o3 and o4-mini** which requires **6 to 140 times higher cost**.
>
> - **Average 8%–12% accuracy gain.** As shown in Table 3 and the **long table in response to reviewer qzNs**, our method improves the F1 score by approximately 20%-50% over the organized prompt baseline across multiple LLMs, and achieves 8%–12% gains over SOTA prompt optimization methods on diverse tasks. Although ProTeGi (GPT-4o mini) achieves slightly higher accuracy than our method on the Swissmetro dataset, our F1 score remains superior (0.498 vs. 0.481). **Considering that ProTeGi requires extensive search and more API calls to find the optimal prompt, our method still outperforms it with an 8.5% higher mean F1 score.**
>
> - **Cross-domain generalization.** Existing RAG-based methods require access to external databases, which are often unavailable in domain-specific tasks. This limits their generalization ability. Our method is designed for scenarios without existing graph databases and uses SCG to support LLM reasoning. SCG requires only minimal expert effort and can be dynamically corrected during optimization, making the approach lightweight and easily adaptable to other tasks. Please see response to **Q2 of Reviewer qzNs** for more detailed experimrnts.
>
> - **Interpretability.** As shown in Figure 5, our unified textual framework enables updating the SCG based on feedback from prediction errors. This offers a direct observation of flaws in expert-provided knowledge and improves transparency in the reasoning process.
>
> - **Co-pilot design.** Our method integrates expert knowledge directly into the LLM reasoning process, enabling the expert to guide and refine the model’s predictions. In turn, the LLM can learn from ground truth feedback and iteratively update the expert-provided knowledge.
>
> >  **Q1: Extra inference cost compared to Zero-shot-CoT or Organized Prompt.**
>
> Thank you for the helpful comment. We evaluated the inference cost using GPT-4o-mini. Each method was tested on the test set and run 5–10 times. The actual expenses were recorded on the OpenAI API platform, and we report the average cost per 100 inferences in USD. The table below compares the accuracy, F1 score, and cost (per 100 examples) across Organized Prompt, Zero-Shot-CoT, and EGO-Prompt. EGO-Prompt incurs approximately 2–3 times higher cost than the Organized Prompt baseline and about 1.5 times higher than Zero-Shot-CoT. However, this additional cost yields notable performance improvements, achieving results comparable to reasoning models such as o4-mini or even o3 which requires 6 to 140 times higher cost.
>
> |Setting|Pandemic Acc|Pandemic F1|Pandemic Cost|TrafficSafe Acc|TrafficSafe F1|TrafficSafe Cost|Swissmetro Acc|Swissmetro F1|Swissmetro Cost|
> |-------|------------|------------|--------------|-------|----|-------|------|--------|---------|
> |Organized Prompt|0.360|0.347|0.016|0.300|0.232|0.020|0.459|0.406|0.019|
> |Zero-Shot-COT|0.370|0.361|0.028|0.280|0.221|0.048|0.494|0.435|0.034|
> |**EGO-Prompt**|**0.410**|**0.399**|**0.042**|**0.380**|**0.333**|**0.078**|**0.506**|**0.498**|**0.048**|
> |o4-mini|0.430|0.418|0.330|0.380|0.334|0.260|0.494|0.456|0.290|
>
> >  **Q2: Graph-based basselines.**
>
> Thank you for the comment. One of our key motivations is to address the gap in domain-specific tasks where no existing structured graph database is available. As Please refer to response to **Q1 of Reviewer #KdBz.**
>
> >  **Q3: Automated ICL methods like AutoCoT**
>
> Thank you for the suggestions. We added two additional Automated ICL baseline methods as follows:
>
> - **Auto-CoT [1]** is an automatic prompting method that generates diverse CoT demonstrations using LLMs. Following its framework, we uniformly sampled 200 training examples for each task and clustered them into 5 groups based on semantic similarity. Within each group, we used the zero-shot "Let's think step by step" prompt to generate CoT reasoning chains for all samples. We then selected the CoT example that produced the correct answer to serve as the demonstration for that cluster.
>
> - **DSPy [2]** is a SOTA LLM programming and prompt optimization framework. We use BootstrappedFewShotRandomSearch optimizer with 10 candidate programs and 5 few-shot examples. The optimizer identifies correct reasoning traces as candidate demonstrations and performs random search over subsets (up to 5 shots) to find the prompt configuration that maximizes task accuracy.
>
> We tested both baselines using GPT-4o-mini as the forward model, and the results are shown in the table below. We observe that the automatic CoT methods perform poorly on our datasets. This aligns with our motivation: LLMs often lack sufficient domain knowledge for these tasks and struggle to learn effective reasoning traces from limited demonstrations, especially given that our input texts are substantially longer than those typically used in CoT benchmark tasks.
>
> |Method|Venue/Journal|Pandemic Acc|Pandemic F1|TrafficSafe Acc|TrafficSafe F1|Swissmetro Acc|Swissmetro F1|Mean F1 (Improvements)|
> |------|--|--|------|----|---|----|----|------|
> |Organized Prompt|—|0.360|0.347|0.300|0.232|0.459|0.406|0.328 (—)|
> |**Auto-CoT [1]**|**ICLR'23**|**0.380**|**0.352**|**0.340**|**0.230**|**0.447**|**0.462**|**0.348 (6.0%)**|
> |**DSPy [2]**|**ICLR'24**|**0.380**|**0.374**|**0.370**|**0.285**|**0.459**|**0.397**|**0.352 (7.2%)**|
> |**EGO-Prompt**|**This work**|**0.410**|**0.399**|**0.380**|**0.333**|**0.506**|**0.498**|**0.410 (24.9%)**|
>
>
> ## References
> [1] Zhang, Z., et al. Automatic chain of thought prompting in large language models. ICLR. 2023.
>
> [2] Khattab, O., et al. Dspy: Compiling declarative language model calls into self-improving pipelines. ICLR, 2024.

---

> > ### Author Response · Authors · 2025-08-06
> >
> > Dear Reviewer UAca,
> >
> > Thank you for your time and effort in reviewing our work. Your feedback is valuable and has helped strengthen our paper. As requested, we have added new experimental results.
> >
> > **Baseline results for Gemini-2.5-flash**.
> > | Method                    | Venue/Journal | Pandemic Acc | Pandemic F1 | TrafficSafe Acc | TrafficSafe F1 | Swissmetro Acc | Swissmetro F1 | Mean F1 (Improvements) |
> > |--------------------------|---------------|---------------|--------------|------------------|----------------|------------------|----------------|-------------------------|
> > | Organized Prompt          | —             | 0.470         | 0.470        | 0.340            | 0.319          | 0.459            | 0.392          | 0.394 (—)               |
> > | **Auto-CoT**| **ICLR'23**   | **0.480**     | **0.472**    | **0.400**        | **0.360**      | **0.494**        | **0.432**      | **0.421 (7.0%)**        |
> > | **DSPy**    | **ICLR'24**   | **0.470**     | **0.460**    | **0.420**        | **0.418**      | **0.506**        | **0.470**      | **0.449 (14.2%)**       |
> > | **EGO-Prompt**            | **This work** | **0.540**     | **0.546**    | **0.430**        | **0.428**      | **0.518**        | **0.499**      | **0.491 (24.7%)**       |
> >
> > We will be more than happy learn from your comments, thoughts, and discuss further if you need!
> >
> > Sincerely,
> >
> > Authors 8465

---

> > > ### Comment · Reviewer_UAca · 2025-08-07
> > >
> > > Thank  the authors' detailed explanations and comprehensive experimental results. This addresses my concerns and confusion.  I will revise my score accordingly.

---

> > > > ### Author Response · Authors · 2025-08-07
> > > >
> > > > We’re glad our rebuttal resolved your concerns. If any additional questions arise, we’re happy to clarify further. We really appreciate your thorough review and constructive feedback!

---

### Official Review · Reviewer_DhWo · 2025-07-01

**Clarity:** 2
**Significance:** 2
**Originality:** 3
**Rating:** 3
**Confidence:** 3

**Summary:**

The work proposes a method for optimising prompts and knowledge graphs to iteratively improve performance on tasks in 3 real-world domains.

**Questions:**

To what extent does the mathematical appearance of Equations 7 & 8 actually add to the rigor of the method?

L517 (Appendix 8.1): Is the Dirac Delta function being used on discrete values?  Doesn't that make it an Indicator function?

**Ethical Concerns:**

["NO or VERY MINOR ethics concerns only"]

**Final Justification:**

Rating remains the same, despite Authors considerable efforts at improving the wording of the paper : These changes make it a much better paper (IMHO).  However, even considering these changes, and rebuttal experiments, the overall system described does not rise above my own mental acceptance hurdle.

**Limitations:**

yes

**Paper Formatting Concerns:**

Care needs to be taken with the formatting of Tables in Section 6.  The widowed line underneath Table 4 is particularly ugly.

Small things:
* L145 "To better leverage ..." -> "To leverage the ... better,"
* L338 'EGO-Promptis' -> 'EGO-Prompt is'

**Quality:**

2

**Strengths And Weaknesses:**

The proposed strategy of both prompt and knowledge graph optimisation is quite interesting, and (guessing from the multiple references to real-world application areas) can be of practical importance.  However, there are aspects of the write-up that detract from the core idea being presented.


'Optimal' is used rather liberally throughout the paper.  While it is true that the Optimiser being proposed aims to produce 'More effective' or 'Improved' versions, suggesting that the results are likely _optimal_ is a stretch (particularly since the end results are <<100% accuracy).  Having the method being called 'Optimiser' is fine, but in several places in the paper the output is called 'optimal' without justification.  For example:

* L118 'enabling an optimal reasoning process' (while 'optimal' is a good objective, your research doesn't 'thereby enable' it).
* L201 "may lead to suboptimal performance" - really means that it didn't work well when you tried it, correct?  Breaking the process into several steps (Algorithm 1) isn't actually achieving 'optimal performance' - it's just a better Algorithm than the previous one...
* L253 'to identify the optimal prompts' -> 'to identify improved prompts'
* Figure 1b : 'Best XYZ' could be 'Improved XYZ' (or Best *so far*)

The language around optimal outputs permeates the paper (i.e. there are many more example than the above).


Around L136 (and equation 3+4) the idea of 'Textual Gradients' is given a mathematical formality which is way beyond the actual notion of derivatives.  At most these should be delta updates (rather than actual formal derivatives).  For instance, L137 : "Chain rule" of these gradients, is really stretching the point.  I realise that this idea can be found in the TextGrad work (now Nature paper), but overloading actual mathematical notation with LLM guesses at text deltas doesn't make it an actually good formalism.


I am not familiar with the datasets being tested against, but it does appear that the results shown may be only marginal improvements.

---

> ### Author Rebuttal · Authors · 2025-07-29
>
> >  **W1: 'Optimal' is used liberally.**
>
> Thank you for the helpful suggestion. We have revised all relevant expressions in the manuscript accordingly. The updates are summarized below:
>
> |**Position**|**Original**|**Revised**|
> |------------|------------|-----------|
> |Page 1, Line 6|Optimal prompts|Improved prompts|
> |Page 2, Line 39|Optimal Domain-adaptive Reasoning|Improved Domain-adaptive Reasoning|
> |Page 2, Line 70-71|Optimal reasoning|Effective reasoning|
> |Page 2, Line 74|Optimal prompts design|Better prompts design|
> |Page 2, Line 76|LLM optimal one-shot reasoning|More effective one-shot reasoning in LLMs|
> |Figure 1|Best/Optimal Prompt/ Graph|Improved Prompt/Graph|
> |Page 4, Line 118|Optimal reasoning process|Better reasoning process|
> |Figure 2|Best Prompt & Graph|Improved Prompt & Graph|
> |Page 6, Line 195|Optimal system prompt|Improved system prompt|
> |Page 6, Line 201|XXX may lead to suboptimal performance|XXX may lead to inferior performance|
> |Page 7, Line 253|Optimal prompts|Better prompts|
>
> >  **W2: The idea of 'Textual Gradients' is given a mathematical formality which is way beyond the actual notion of derivatives.**
>
> Thank you for the constructive suggestion.
>
> - **We fully agree with your comment and have revised the notational system.** In our original manuscript, we adopted the formulation used in the TextGrad paper (see Figure 2 of [1]), where derivatives and chain rules are expressed using the partial derivation [1]. However, we fully understand your concern regarding the potential over-formalization and agree that such notation may overstate the actual mathematical rigor behind textual feedback. In fact, this concern resonated with us as well when we first encountered those equations. In light of this, we have revised our equations to more appropriately reflect delta-style updates rather than formal gradients, and we have added clarifying explanations to reduce the risk of misinterpretation.
>
> | **Eq. No.** | **Original** | **Revised** |
> |------------|--------------|----------------------------|
> | **3** | $\nabla\_{\mathcal{P}\_{\text{sys}}} \mathcal{L}(\hat{y}\_i\, y\_i) = \frac{\partial \mathcal{L}(\hat{y}\_i\, y\_i)}{\partial \mathcal{P}\_{\text{sys}}} = \mathcal{M}\_B(x\_i\, y\_i\, \mathcal{P}\_{\text{sys}}\, \mathcal{L}(\hat{y}\_i\, y\_i))$ | $ \Delta \mathcal{L}\_i = \mathcal{M}\_B(\hat{y\_i}\, \mathcal{L}\_i)\, \Delta P\_{\text{sys}}=\mathcal{M}\_B( P\_{\text{sys}}\, \hat{y\_i}\, \Delta \mathcal{L}\_i ) $ |
> | **7** | $\nabla\_{\mathcal{P}\_{\text{sys}}} \mathcal{L}\_i = \frac{\partial \mathcal{L}\_i}{\partial \hat{y}\_i} \cdot \frac{\partial \hat{y}\_i}{\partial \mathcal{P}\_{\text{sys}}}$ | Removed, refer to Eq. (3) |
> | **8** | $\nabla\_{\mathcal{G}} \mathcal{L}\_i = \nabla_{z^{\*}\_i} \mathcal{L}\_i \cdot \frac{\partial z^{\*}\_i}{\partial \mathcal{G}}\, \quad \nabla\_{\mathcal{P}\_{\text{cau}}} \mathcal{L}\_i = \nabla_{z^{\!*}\_i} \mathcal{L}\_i \cdot \frac{\partial z^{\*}\_i}{\partial \mathcal{P}\_{\text{cau}}}$ | $\Delta P\_{\text{cau}}=\mathcal{M}\_B( P\_{\text{cau}}, z^{\*}_i\, \Delta z^{\*}\_i)\, \Delta \mathcal{G}=\mathcal{M}\_B(\mathcal{G}\, z^{\*}\_i\, \Delta z^{\*}\_i)$\, where $\Delta z^{\*}\_i=\mathcal{M}\_B(\hat{y}_i,z^{\*}\_i\,\Delta \mathcal{L}\_i )$ |
>
> - **Textual Gradient is only used as the iteration optimizer, like Adam. We also use it as a baseline method.** Our core idea is the allignment of expert knowledge with LLM reasoning process. By introducing the SCG, this focus becomes aligning the graph-based information with the textual information and jointly optimizing both the graph and the prompt to better guide the model’s reasoning. To achieve this, we use the instance-spcific guidance to describe the graph as a better format to be integrated in textual prompt, and develop the iterative optimization method to co-optimize the SCG and system prompt. This approach allows the SCG to be corrected and completed during training, while the system prompt is simultaneously adapted to effectively leverage the SCG for improved performance. **In this entire process, Textual Gradient is simply used as a black-box optimizer**. We included Section 3 to briefly introduce the core idea of Textual Gradient, particularly because many readers may be unfamiliar with it and we have properly cited the original TextGrad work.
>
>
> > **W3: The results shown may be only marginal improvements.**
>
> Thank you for your comment. We understand your concern regarding the performance improvements. However, given that the performance of LLMs is generally constrained by **scaling laws**, we would like to clarify that our contributions extend beyond prediction accuracy and also include improvements in **cost-efficiency, cross-domain generalization, interpretability, and the co-pilot design**.
>
> - **High accuracy.**As shown in Table 3 and the **long table in response to reviewer qzNs**, our method improves the F1 score by approximately 20%-50% over the organized prompt baseline across multiple LLMs, and achieves 8%–12% gains over SOTA prompt optimization methods on diverse tasks.
>
> - **Cost-efficiency.** As shown in Figure 4 and Table 7, we make **GPT-4o-mini comparable with reasoning models like o3 and o4-mini** which requires **6 to 140 times higher cost**.
>
> - **Cross-domain generalization.** Existing RAG-based methods require access to external databases, which are often unavailable in domain-specific tasks. This limits their generalization ability. Our method is designed for scenarios without existing graph databases and uses SCG to support LLM reasoning. SCG requires only minimal expert effort and can be dynamically corrected during optimization, making the approach lightweight and easily adaptable to other tasks. Please see response to **Q2 of Reviewer qzNs** for more detailed experimrnts.
>
> - **Interpretability.** As shown in Figure 5, our unified textual framework enables updating the SCG based on feedback from prediction errors. This offers a direct observation of flaws in expert-provided knowledge and improves transparency in the reasoning process.
>
> - **Co-pilot design.** Our method integrates expert knowledge directly into the LLM reasoning process, enabling the expert to guide and refine the model’s predictions. In turn, the LLM can learn from ground truth feedback and iteratively update the expert-provided knowledge. This co-pilot mechanism fosters a more interactive and adaptive decision-making process in domain-specific tasks.
>
> >  **Q1: How Equations 7 \& 8 actually add to the rigor of the method.**
>
> Thank you for the comment. We added Equations 7 and 8 with the following considerations:
>
> - **Explicitly formulate the update process of the variables.** Since our method involves two LLMs and updates three key components (the system prompt, the causal system prompt, and the SCG), we aimed to clearly specify how each variable is updated and which information it depends on. For instance, Equation 8 illustrates that the SCG is updated through the textual gradient of the reasoning guidance, which is approximated through a chain of feedback signals propagated from the prediction error back to the guidance itself. We acknowledge that the original equation may have appeared somewhat difficult to follow, but we believe it is important for clarifying the update process. Based on your suggestion in W2, we have revised the gradient notation using delta-style updates, which improves readability.
>
> - **Readers may not familiar with the chain rule in Textual Gradient.** As noted in W2, some readers may not be familiar with the concept of Textual Gradient and its underlying propagation process. We believe that Equations 7 and 8 can help facilitate understanding by providing a structured view of how feedback signals are propagated through different components of our framework.
>
> >  **Q2: Dirac Delta function being used on discrete values make it an Indicator function.**
>
> Thank you for your careful observation. You are correct that the Dirac delta function is technically defined over continuous domains, whereas in our formulation, the reasoning guidance is discrete (i.e., a sequence of textual statements, though from a vast space).
>
> To reflect this more accurately in the discrete setting, we have revised the expression to use an indicator function:
>
> $$p(y \mid x\, \mathcal{G}) = \sum\_{z} p(y \mid x\, z)\mathbb{I}\left[z = z^{\*}(x\, \mathcal{G})\right] = p\left(y \mid x\, z^{\*}(x\, \mathcal{G})\right)$$
>
>
>
> >  **Paper Formatting Concerns: Tables in Section 6, grammer corrections.**
>
> Thank you for your suggestion. We have now revised the layout and corrected the grammatical errors accordingly.
>
> ## References
> [1] Mert Yuksekgonul, Federico Bianchi, Joseph Boen, Sheng Liu, Pan Lu, Zhi Huang, Carlos Guestrin, and James Zou. Optimizing generative ai by backpropagating language model feedback. Nature, 639(8055):609–616, 2025.

---

> > ### Comment · Reviewer_DhWo · 2025-08-04
> >
> > W1: These changes make the paper more accurate
> >
> > W2: These changes make the algorithm clearer.  The ideas of 'Textual Gradients' are interesting, but the partial derivative notation is a step too far, IMHO.
> >
> > W3: With all due respect, calculating improvement percentages as a multiple of the baseline accuracy is something that (apparently) even embarrassed the authors when looking at the Qwen results.
> >
> > Thank you for your rebuttal changes.  However, at present I am not inclined to increase my score.

---

> > > ### Author Response · Authors · 2025-08-04
> > >
> > > **1. W1, W2.**
> > >
> > > Thank you for your expertise. Your review and comments has strengthened our work.
> > >
> > > **2. W3.**
> > >
> > > - Reporting relative improvement is a widely used approach to present performance gains. For example, in the ResNet paper [1], the authors include the following table and state that “we obtain a 28% relative improvement on the COCO object detection dataset” in their abstract. We believe this type of report can save both reader and reviewer’s time. We hope you'll reconsider the advantages and convenience this approach can bring in.
> > >
> > > **Table 8 in ResNet paper [1]:**
> > > | Metric     | mAP@.5 | mAP@[.5, .95] |
> > > |------------|--------|---------------|
> > > | VGG-16     | 41.5   | 21.2          |
> > > | ResNet-101 | 48.4   | 27.2          |
> > >
> > > - The large improvements observed for small models (119.5% for the 1.7B model and 200.1% for the 8B model) occur because these models fail to generate structured outputs under the initial organized prompt. With EGO-Prompt, however, they are able to produce more structured predictions and follow consistent reasoning paths. We exclude them as our improvements as shown in response to Q1 of Reviewer qzNs.
> > >
> > > - **Comparing with existing SOTA methods, our EGO Prompt achieve 8%-12% improvements**. We want to highlight the results for Qwen clearly demonstrate our method’s ability to improve a model’s reasoning capacity in domain-specific tasks:
> > >
> > >   - Before EGO-Prompt, the models did not follow the expected scaling law (e.g., the mean F1 of the 14B model, 0.279, was slightly higher than that of the 32B model, 0.278). Assuming similar training data, this performance gap likely stems from differences in their reasoning capability. The breakdown of the scaling law here suggests that the models lack sufficient reasoning ability for domain-specific tasks.
> > >
> > >   - After EGO-Prompt, performance increases consistently with larger parameter counts. This suggests that our method helps models to approach the upper bound of their reasoning capability given the existing knowledge and conditions.
> > >
> > > Please let us know if you have any further technical questions or concerns regarding our method and results; we would be happy to discuss them! Thanks again!
> > >
> > > ## References
> > > [1] He, Kaiming, et al. "Deep residual learning for image recognition." Proceedings of the IEEE conference on computer vision and pattern recognition. 2016.

---

> > > ### Author Response · Authors · 2025-08-08
> > > **Updates on GPT-5-mini Results**
> > >
> > > Dear Reviewer DhWo,
> > >
> > > We appreciate your suggestions on paper writing and mathematical formulation, which have clearly made the paper more readable and accurate. We also value your attention to the potential improvements of our method. In response, we have added a series of new baselines to better demonstrate the advantages of our approach.
> > >
> > > Following the release of OpenAI’s new GPT-5 series models, we conducted additional experiments to evaluate our method on the GPT-5-mini (minimal reasoning) model. The results, provided in our **rebuttal to Reviewer qzNs**, show that our method remains effective for the latest lightweight models.
> > >
> > > If you have any further concerns, we would be happy to discuss them.
> > >
> > > Authors, 8465

---

### Official Review · Reviewer_KdBz · 2025-07-02

**Clarity:** 2
**Significance:** 2
**Originality:** 2
**Rating:** 4
**Confidence:** 3

**Summary:**

The paper introduces EGO-Prompt, a framework designed to optimize prompt design and reasoning for LLMs in domain-specific tasks by incorporating expert-defined causal graphs. It begins with imperfect Semantic Causal Graphs (SCGs) and iteratively refines both the SCG and system prompts using textual gradients. This method produces instance-specific reasoning guidance, enhancing model accuracy without requiring fine-tuning. Applied to public health, transportation, and behavioral datasets, EGO-Prompt outperforms prior methods with F1 score improvements of up to 12%, while also reducing inference costs.

**Questions:**

In addition to addressing the concerns raised under the weaknesses section, we encourage the authors to clarify the following questions:

1）	Could the authors elaborate on the distinction between “expert knowledge” and “database knowledge” as described in the introduction and illustrated in Figure 1(a)? Specifically, why are RAG methods considered unsuitable for utilizing expert knowledge in this context?

2）	How do the authors ensure that prompting with natural language effectively captures the complex, higher-order relationships inherent in graph structures? A discussion on the limitations and representational fidelity of such text-based graph encodings would be beneficial.

**Ethical Concerns:**

["NO or VERY MINOR ethics concerns only"]

**Final Justification:**

My major concerns have been addressed. I have raised the score.

**Limitations:**

Yes

**Paper Formatting Concerns:**

No formatting concerns.

**Quality:**

2

**Strengths And Weaknesses:**

Strengths:

1)	The paper presents a well-structured and comprehensible pipeline for integrating domain knowledge into LLM reasoning, making the methodology easy to follow and replicate.

2)	The use of SCGs offers a structured and flexible way to encode and evolve domain knowledge, enhancing the model’s ability to reason effectively in complex, real-world settings.

3)	Through detailed ablation studies, the paper validates the individual impact of its core components—such as instance-specific reasoning guidance and iterative optimization—on overall performance.

Weaknesses:

1)	The paper does not compare EGO-Prompt against reinforcement learning-based prompt optimization methods (e.g., [1]), which are important alternatives in the field and could provide valuable benchmarks.

2)	Although six challenges are listed, their interrelationships and how the proposed method specifically addresses each are not clearly articulated. For instance, it is unclear how using a graph-based structure helps resolve the “task evolutionary” challenge or how the components of EGO-Prompt systematically map to the stated problems.

3)	The claim that EGO-Prompt reduces human involvement lacks empirical support. Since the approach still relies on expert-created prompts and initial causal graphs, a quantitative analysis—such as testing performance under varying degrees of SCG completeness—would clarify the true extent of human dependency.

4)	Some experimental outcomes are counterintuitive. For example, in Table 5, using the graph seems to yield worse performance on the Swissmetro dataset compared to not using it (as seen in Table 4). The paper does not provide sufficient explanation for such discrepancies, which raises concerns about the method’s robustness across tasks.

Reference

[1] Tempera: Test-time prompting via reinforcement learning.

---

> ### Author Rebuttal · Authors · 2025-07-29
>
> >  **W1: The diference with RL-based methods.**
>
> Thanks! We understand the question and agree that comparing with existing RL methods would be interesting. However, we believe such a comparison would not be fair or meaningful, as our goal differs fundamentally from RL-based prompt optimization. We aim to develop a generalized prompt optimization framework applicable for all LLMs in domain-specific tasks, rather than optimizing prompts for individual models with RL on each domain task.
>
> - **Distinction of prompt optimization paradigms.** Prompt optimization methods can be broadly categorized into two types: (1) continuous optimization, which directly updates model parameters; and (2) discrete optimization, which searches for improved prompts in natural language form. Most RL-based methods fall into the continuous category and are designed for open-source models.
>
> - **RL optimization may not be integrable or generalizable in closed-source LLMs. Our motivation is to leverage all LLMs for better domain-specific task adaptations (e.g., public health, transportation).** The end users are often domain experts with limited access to computational resources. Recent RL-based methods for prompt optimization typically require local fine-tuning or policy learning with open-source models [1,2], which conflicts with our motivation of lightweight and broadly accessible solutions for domain experts. In practice, if servers, model weights, and labeled data are available (always true in domain specific tasks), fine-tuning can be a stronger alternative.
>
> - **We add more baselines for both close- and open-source LLMs.**. Please see response to **Q3 of Reviewer UAca**.
>
> We have revised the discussion to clarify our motivation and distinguish our approach from RL-based work [1,2] from scope and targets.
>
> >  **W2: How the proposed method addresses challenges.**
>
> Thanks! We emphasized three challenges for seeking the potentially best prompt and reasoning procedure for domain-specific tasks without fine-tuning.
> - Domain-knowledge Adaptation. We encode expert knowledge using a SCG that captures abstract causal relationships.
> - Optimal Domain-adaptive Reasoning. The SCG guides the reasoning process, improving efficiency and performance.
> - Task Evolutionary. We dynamically refine the SCG using ground-truth data, which in turn updates expert knowledge through a reverse feedback loop.
>
> We also overcome key limitations of exsiting RAG-based methods:
> - Partially Observed Domain Graphs. We avoid reliance on external databases by using lightweight, domain-specific SCGs.
> - Graph Priors Evolution. Please see W4 for details.
> - Optimal Graphical-guided Reasoning Process. Please see Q2 for details.
>
> We have revised the introduction section of the manuscript to provide a simpler and more direct description.
>
> >  **W3: Quantitative analysis of human dependency.**
>
> Thanks! To better understand the impact of human-provided SCGs, we conducted additional experiments under two settings:
>
> - **Reversed SCG.** To test sensitivity to incorrect causal structures, we reversed all edges in the expert-designed SCGs used in the main experiments. This allows us to compare performance differences between mostly correct and mostly incorrect SCGs.
>
> - **Null SCG.** We removed the initial SCG and let the model construct it during optimization. This tests whether providing a good initial SCG improves performance and whether SCG can be constructed automatically.
>
> |Setting        |EGO Opt.|Pandemic F1|TrafficSafe F1|Swissmetro F1|
> |---------------|--------------|------------|----------------|---------------|
> |Organized Prompt||0.347|0.232|0.406|
> |Reversed SCG   ||0.319|0.160|0.277|
> |Reversed SCG   |✓|0.303|0.260|0.457|
> |Null SCG       ||0.330|0.224|0.295|
> |Null SCG       |✓|0.372|0.317|0.470|
> |EGO-Prompt     |✓|**0.399**|**0.333**|**0.498**|
>
> From the experiment results, we found:
>
> - **Reversed SCG degrades reasoning performance.** Without optimization, reversing all causal edges introduces misleading priors and reduces performance.
> - **Reversed SCG damages prompt optimization.** On the Pandemic dataset, performance worsens even after optimization (F1 drops from 0.319 to 0.303), suggesting that incorrect prior knowledge can misguide the optimization process.
> - **EGO-Prompt can recover from a null SCG while carefully designed SCG is better.** A null SCG causes a slight initial drop (e.g., Pandemic F1: 0.347 → 0.330). This is expected, as the model receives no causal guidance at the start. After optimization, the model can construct a meaningful SCG, resulting in performance recovery.
> - **Correct SCG can yield better performance.** The expert-designed SCG consistently leads to the highest performance, both before and after optimization. In contrast, reversed SCGs degrade performance and obstruct optimization, while null SCGs hinder initial reasoning but allow recovery through learning.
>
> >  **W4: Lack explaination to Table 4&5.**
>
> Thanks! These experimental results directly link to the motivation of our work.
>
> - **Table 4.** Table 4 shows two results: (1) using a single model to integrate the system prompt and SCG; and (2) our proposed method, which uses instance-specific reasoning guidance and a separate model to translate the SCG into textual guidance. The results show that our model outperformed than the single model design. As both experimrnts incorporate SCG, this results are not related to the performance degration issue brought by graph.
> - **Why graph degrade the performance in Table 5.** In Table 5, we observe that fixing either the SCG or the system prompt leads to degraded performance, while jointly optimizing both yields the best results. This is due to potential inaccuracies in expert-designed SCGs and misalignment when the prompt isn't adapted to the SCG.
> - **How SCG handle with the flawed graph.** We focus on aligning expert knowledge with the LLM’s reasoning process. By introducing the SCG, this focus becomes **aligning the graph-based information with the textual information** and **jointly optimizing both the graph and the prompt** to better guide the reasoning. So, we use the instance-spcific guidance to describe the graph as a better format to be integrated in textual prompt, and develop the iterative optimization method to co-optimize the SCG and system prompt. This approach allows the SCG to be corrected during training, while the system prompt is simultaneously adapted to effectively leverage the SCG for better performance.
>
> >  **Q1: Distinction between “expert knowledge” and “database knowledge”.**
>
> - **Graph database encodes entity-level factual knowledge**, e.g., ("Crash #1234", "occurredOn", "Highway-99"). This brings knowledge, but can not guide better reasoning. In contrast, **expert knowledge typically appears as conditional rules, heuristics, and mental models that operate under uncertainty,** focusing on the reasoning process rather than static facts. For example, “If the driver is fatigued and the road is slippery, the crash risk increases markedly”. Such judgments require abstract causal relations and contextual understanding that are difficult to encode as static database entries and cannot be reliably retrieved by RAG-based methods.
> - **SCG is an effective tool for representing expert knowledge.** SCG serves as a powerful framework to capture expert knowledge by combining a structured graph that encodes causal relationships with textual descriptions that convey factual information. Moreover, SCG can be losslessly integrated into the reasoning process of LLMs (see Equation (1) and Appendix 8.1) and dynamically updated through prompts using the Textual Gradient method.
> - **Why RAG is unsuitable here.** RAG methods rely on retrieving discrete, surface-level information chunks from a knowledge base. However, in our setting:
>
>     - There is **limited existing large-scale domain-specific knowledge graph database**. This is the common settings in real-world applications.
>     - Even if such a database existed, **retrieving knowledge fragments is insufficient to guide structured reasoning** across multiple interacting factors, as required in our tasks. SCG goes beyond retrieval and external knowledge supplementary and instead structures the model’s reasoning path.
>
> >  **Q2: Can EGO-Prompt capture the complex, higher-order relationships in graph?**
>
> Thanks! We fully agree that this is a challenging but insightful idea. SCG provides a compact representation of expert reasoning by encoding abstract causal relationships between key concepts (see Q1). While graphs capture structural dependencies, they often lack the contextual richness needed for nuanced decision-making. For example, a graph might indicate “alcohol increases crash risk,” but not capture conditions like “young drivers at night in rural areas face higher risk.”
>
> **Instance-specific reasoning guidance bridges the gap between structured graph and natural language.** Graph-structured information, such as SCGs, is not directly compatible with LLM inputs in its raw form. To better integrate graph information into natural language prompts, we introduce a causal description model (see Figure 2) that aligns SCGs with natural language. By satisfying the Conditional Independence and Deterministic Guidance conditions (see Section 8.1), the model produces lossless textual descriptions that better preserve the high-order dependencies encoded in the graph while making them accessible to language models. Our experiments (Table 4) show that this design substantially improves performance over directly feeding graphs into the LLMs — **so the answer is yes.** We are also working on improving causal graph representation through symbolic graph representations.
>
> ## References
> [1] Zhang, T., , et al. Tempera: Test-time prompting via reinforcement learning. ICLR. 2023
>
> [2] Kwon, M.,, et al. StablePrompt: automatic prompt tuning using reinforcement learning for large language models. EMNLP. 2024

---

> > ### Comment · Reviewer_KdBz · 2025-08-05
> >
> > Thank the authors for their detailed response. However, I still have a few lingering concerns:
> > 1.	On W1: My understanding is that approaches such as [1] perform prompt refinement directly in natural language without parameter updates, which places them under the same broad category of the proposed method. This seems to contrast with the authors' rebuttal, which emphasized that recent RL-based methods typically require local fine-tuning or policy learning.
> > 2.	On W3: While I appreciate the inclusion of extreme baselines like reversed and null SCGs, I still believe that an ablation examining the effect of gradually removing correct SCG edges (e.g., at different masking ratios) would offer more actionable insights into how sensitive the method is to partial human supervision.
> >
> > [1] Tempera: Test-time prompting via reinforcement learning.

---

> ### Author Response · Authors · 2025-08-06
>
> We sincerely appreciate your time and effort. Your feedback is constrctive has undoubtedly helped strengthen our work. Below, we respond to the concerns you raised.
>
> **W1.**
>
> As mentioned, recent RL-based methods for prompt optimization typically require either local fine-tuning or policy learning using open-source models. These approaches can be broadly categorized into two types: (1) those that directly update the parameters of the LLMs, and (2) those that update the parameters of a separate policy model. The paper you referred to falls into the second category, where the LLM itself is not directly updated.
>
> However, this approach still conflicts with our settings for the following reasons:
>
> - **It requires access to the LLM’s logits output, which is not available from commercial models.** The reward function in this paper relies on logits to compute reward signals. While recent research has explored ways to eliminate this requirement, these methods still face another major limitation:
>
> - **They involve training a policy model that must be reasonably strong, ideally not much weaker than the target LLM.** In the referenced paper [1], the authors use the encoder layers of GPT (with unspecified parameter size) as the policy network to optimize prompts for RoBERTa-large (approximately 0.35B parameters). In another paper we cited in the rebuttal [2], both the target and policy models are of comparable size, typically around 7B or more. In summary, RL-based methods generally require updating a policy model of similar scale to the target model, which contradicts our goal of avoiding costly training and heavy infrastructure.
>
> |**Method Type**|**Target Model Update**|**Policy Model Update**|**Target Model Logits Requirement**|**User Expertise Requirement**|
> |---|---|---|---|---|
> |Direct LLM Update (Type 1)|Yes|No|Required|High|
> |Policy-guided RL (Type 2) [1]|No|Yes|Required|High|
> |Black-box RL (Recent) [2]|No|Yes|Not Required|Medium to High|
> |**Our Method**|No|No|Not Required|Low|
>
> **W3.**
>
> We have added new experiments by randomly removing one-third and two-thirds of the causal connections in the SCG:
>
> | **SCG Setting** | **Pandemic Acc** | **Pandemic F1** | **TrafficSafe Acc** | **TrafficSafe F1** | **Swissmetro Acc** | **Swissmetro F1** | **Mean F1** |
> |----------------|------------------|------------------|----------------------|---------------------|---------------------|--------------------|-------------|
> | Reversed SCG   | 0.310            | 0.303            | 0.290                | 0.260               | 0.482               | 0.457              | 0.350       |
> | Empty SCG      | 0.370            | 0.372            | 0.330                | 0.317               | 0.506               | 0.470              | 0.394       |
> | 33% SCG        | 0.390            | 0.389            | 0.300                | 0.270               | 0.459               | 0.462              | 0.378       |
> | 66% SCG        | 0.400            | 0.387            | 0.310                | 0.314               | 0.506               | 0.493              | 0.402       |
> | Full SCG       | 0.410            | 0.399            | 0.380                | 0.333               | 0.506               | 0.498              | 0.421       |
>
>
> We have the following findings:
>
> - **Randomly removing causal edges can impair the model’s reasoning capability.** The initial SCG represents a complete reasoning path. Although it may not be entirely correct, it maintains structural integrity and connected as a DAG. Random removal of edges disrupts this structure, leading to degraded performance. For example, the 33% SCG setting yields a lower mean F1 score than the Empty SCG, indicating that a partially broken causal graph may be worse than having no guidance at all.
>
> - **Greater completeness in the SCG supports better reasoning.** The 66% SCG setting achieves performance close to that of the full SCG, suggesting that more complete causal guidance helps the model reason more effectively.
>
> - **If a user is unsure about the completeness or correctness of the SCG, it is advisable to start with an empty SCG and iteratively refine it.** This allows gradual construction of a reliable causal graph without risking the adverse effects of incorrect or incomplete edges.
>
> As the time is limited, we conducted this experiment to the best of our ability. If you have any further suggestions or comments, we are happy to hear and discuss them. Thank you again for your feedback!
>
>
>
> Sincerely，
>
> The Authors of 8465
>
> ## References
> [1] Zhang, T., Wang, X., Zhou, D., Schuurmans, D., & Gonzalez, J. E.. Tempera: Test-time prompting via reinforcement learning. arXiv preprint arXiv:2211.11890. 2022
>
> [2] Kwon, M., Kim, G., Kim, J., Lee, H., & Kim, J. StablePrompt: automatic prompt tuning using reinforcement learning for large language models. arXiv preprint arXiv:2410.07652. 2024

---

> ### Author Response · Authors · 2025-08-08
> **Updates on GPT-5-mini Results**
>
> Dear Reviewer KdBz,
>
> Thank you for the constructive comments. Your suggestions, such as the quantitative analysis of human dependency, have substantially improved the quality of our paper. Your comment on the RL-based method has also helped us draw a clearer distinction between our work and other related methods.
>
> Following the release of OpenAI’s new GPT-5 series models, we conducted additional experiments to evaluate our method on the GPT-5-mini (minimal reasoning) model. The results, provided in our **rebuttal to Reviewer qzNs**, show that our method remains effective for the latest lightweight models.
>
> If you have any further concerns, we would be happy to discuss them.
>
> Authors, 8465

---

### Note · Authors · 2025-08-12

We thank all reviewers and the AC for thoughtful feedback that strengthened our work. Below, we summarize the final paper, clarifications, and contributions.

## 1. Core Contribution
EGO-Prompt’s primary contributions are:

1. **Effective and Efficient Domain-specif Reasoning.** Our method improves F1 by 20%–50% over the organized prompt baseline across multiple LLMs, making GPT-4o-mini comparable to reasoning models like o3 and o4-mini at 6x–140x lower cost.

2. **Lossless Expert Knowledge Integration and Evolution.** By leveraging the SCG and instance-specific reasoning guidance, expert knowledge is losslessly incorporated into the LLM’s reasoning process. The SCG is updated to refine flawed or incorrect knowledge, enabling a self-improving feedback loop.

## 2. Addressed Concerns
- Quantitative analysis of human dependency. We added experiments under (1) reversed SCG, (2) empty SCG, and (3) partial SCGs with 33% and 66% edges retained, to assess human influence on SCG quality (W3 of KdBz, W1 of qzNs).
- Baselines. We included two new Automated ICL baelines: Auto-CoT and DSPy (Q3 of UAca). We clarified how our approach differs from RL-based methods (W1 of KdBz).
- Gneralization. We add GPT-5-mini results. We also extended evaluation to open-source models including Qwen3 (1.7B, 8B, 14B, 32B), DeepSeek-V3, LLaMA-3.3 (8B, 70B), and LLaMA-4 (Q1 of qzNs). We also evaluated generalization under different dataset distributions (Q2 of qzNs).
- Inference and optimization cost. Addressed in W2 and Q1 of Reviewer UAca.
- Writing and formulation. Revised based on Reviewer DhWo’s comments.

## 3. Broader Impact

EGO-Prompt removes the requirement for a graph database in existing RAG methods, which is typically unavailable in real-world tasks. Instead, expert knowledge is captured through the SCG and aligned with the LLM’s reasoning process, enabling more flexible and robust domain-specific reasoning. Beyond prompt optimization, EGO-Prompt can also:

- Automated Domain Knowledge Extraction and Refinement. The SCG can be iteratively refined using ground truth, and even constructed from scratch (empty SCG), enabling knowledge discovery in new domains.

- Co-polit real-world decision-making. Experts provide knowledge and guidance to LLMs, while LLMs, in turn, correct and enhance expert knowledge, creating a dynamic and mutually beneficial feedback loop. This is especially valuable in scenarios where new information emerges (e.g., a new variant of a pandemic virus).

---

### Decision · Program_Chairs · 2025-09-17

**Decision:**

Accept (poster)

**Comment:**

This paper introduces EGO-Prompt, a novel framework for improving the reasoning capabilities of Large Language Models (LLMs) on domain-specific tasks. The core contribution is a method that iteratively co-optimizes both a human-initialized Semantic Causal Graph (SCG) and the system prompts used to guide the LLM.

The core idea of jointly optimizing a knowledge graph and system prompts in an evolutionary manner is novel, the paper is easy to follow, and the results are strong. There were initial concerns raised by the reviewers, but authors managed to address most concerns in the rebuttal phase. Therefore, we have decided to accept this paper.